# SPROUT: Training-free Nuclear Instance Segmentation with Automatic Prompting

## Abstract

Nuclear instance segmentation is a cornerstone task in digital pathology, with broad potential to drive clinical decision-making and accelerate therapeutic discovery. Recent advances in large vision foundation models have shown promise for zero-shot segmentation in biomedical domains. However, most efforts in pathology still rely on pre-trained vision models through fine-tuning or adapter modules. These approaches demand costly annotations and heavy computation, leaving efficient training-free methods largely unexplored. To this end, we propose **SPROUT**, a simple yet effective framework for annotation-free prompting. Specifically, we leverage histology-informed stain priors to construct slide-specific references for mitigating domain gaps and instantiate a prototype-guided partial optimal transport scheme to progressively refine nuclear representations. In addition, we embed high-quality positive and negative prompts into the Segment Anything Model (SAM) *without any fine-tuning*. Extensive experiments across multiple histopathology benchmark datasets demonstrate that SPROUT achieves competitive performance while requiring neither annotations nor retraining. These results establish SPROUT as a scalable, training-free solution for nuclear instance segmentation in pathology. Our codes are available at here.

## 1 Introduction

Nuclear instance segmentation delineates individual nucleus for systematic downstream analysis (Caicedo et al., 2019; Greenwald et al., 2022; Gupta et al., 2023) and advance cancer prognosis, diagnosis, and treatment (Madabhushi & Lee, 2016; Lu et al., 2018; Pinckaers et al., 2021). In histopathology, cellular structure visualizations are most commonly obtained from hematoxylin and eosin (H&E) staining (Vahadane et al., 2016). Hematoxylin highlights nuclei in dark blue or purple and eosin stains cytoplasm and extracellular components in pink for separation (Ruifrok et al., 2001). However, the intrinsic properties of H&E pathology images pose unique challenges for instance segmentation. First, the narrow color spectrum and staining variability limit robust visual cues. Second, a single patch can contain thousands of densely packed nuclei with weak boundaries. Third, pixel-wise annotations are scarce, costly, and labor-intensive for pathologists.

To address these challenges, numerous specialized nuclear segmentation networks have been explored under varying levels of supervision, including fully-supervised (Graham et al., 2019; Qu et al., 2019; He et al., 2021b; Chen et al., 2023a; He et al., 2023), semi-supervised (Zhou et al., 2020; Wu et al., 2022; Jin et al., 2022), weakly-supervised (Zhao & Yin, 2020; Nishimura et al., 2021; Liu et al., 2022), and self-supervised (Sahasrabudhe et al., 2020; Xie et al., 2020). Nevertheless, their performance is often limited when facing distribution shifts, varying annotation protocols, and restricted training data (Pachitariu & Stringer, 2022). Together, these limitations underscore the pressing need for generalizable and robust approaches to nuclear instance segmentation.

Vision foundation models have marked a turning point in image segmentation. Segmentation Anything Model (SAM) (Kirillov et al., 2023) achieves robust zero-shot, class-agnostic segmentation ability by leveraging large-scale training on the vast SA-1B datasets. Building on SAM, subsequent work has explored fine-tuning (Zhang et al., 2024c; Ma et al., 2024; Peng et al., 2024; Archit et al., 2025) and adapter-based strategies (Chen et al., 2023b; Na et al., 2024; Cheng et al., 2024; Chen et al., 2025a) to accommodate SAM to task-specific medical objectives. However, the need for substantial annotations and resource-intensive training constraints their practicality in pathology.

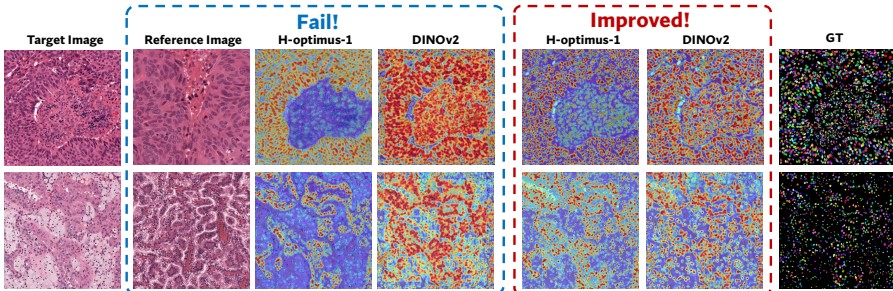

Figure 1: **Comparison of one-shot and proposed self-reference strategies in feature extraction.** One-shot (**blue box**) fails to capture precise and diverse nuclei, even with similar pairs or backbones trained on natural and pathology images. Instead, our self-reference approach (**red box**) leverages high-confidence regions within the image to extract more robust features for similarity guidance.

An appealing alternative is to transform feature correspondences between the annotated references and targeted images into actionable prompts (Liu et al., 2024; Zhang et al., 2024a; Liu et al., 2025) without supervision or retraining. One might reasonably expect obtaining high-quality prompts from external backbones trained on natural or pathological domains, such as DINOv2 (Oquab et al., 2024) and H-optimus-1 (Bioptimus, 2025). Yet the distinct properties of pathological images hinder such direct transfer. As illustrated in Figure 1, even when target and reference pairs are carefully matched in color, nuclear size, and spatial distribution, backbone models exhibit characteristic failures. Feature extractors trained on natural images can over-amplify the nuclear regions, while pathology-trained ones still struggle to capture subtle, heterogeneous cell-level features. Unlike natural images with a limited number of salient objects occupying large portions, nuclei are harder to capture reliably for fine-grained structures that may consist of only a few thousand pixels. Moreover, few-shot strategies are often impractical in pathology. Variations in staining, cellular density, and local morphology preclude the establishment of appropriate references or consistent image matching. Consequently, their unstable performance across references is unsuitable for practical deployment.

In this work, we address these challenges by introducing **SPROUT** (**S**tain **P**riors with p**R**ototypical partial **O**ptimal transport for **U**nlabeled promp**T**ing), an automatic self-reference prompting framework that guides SAM for nuclear instance segmentation *without any training*. Specifically, we leverage the biochemical affinity of H&E staining to generate slide-specific foreground and background *self-reference* regions for feature similarity identification. Such a strategy is surprisingly effective as the references calibrate the image-specific feature activations for more precise and complete nuclear delineation, as illustrated in Figure 1. To achieve exact coverage of the cells and mitigate ambiguity, we propose **POT-Scan**, a principled partial optimal transport scheme built on feature–prototype similarity mapping. POT-Scan progressively refines class activation maps through comprehensive prototype encapsulation. In addition, we incorporate biological priors into a novel containment-aware Non-Maximum Suppression (NMS) strategy for SAM prediction refinement. We empirically validate our SPROUT across three public benchmark datasets, *i.e.*, MoNuSeg (Kumar et al., 2017), CPM17 (Vu et al., 2019), and TNBC (Naylor et al., 2018), and demonstrate remarkable performance compared with SAM-based, fully-supervised, and weakly-supervised counterparts while remaining computationally efficient. Our main contributions are summarized as follows:

- To the best of our knowledge, SPROUT is the first fully training-free framework for nuclear instance segmentation in H&E pathology images without annotations. By addressing the limitations of reference-based methods, we introduce a novel self-reference mechanism that offers a lightweight yet generalizable solution to domain gaps.

- We propose POT-Scan, a principled scheme with theoretical guarantees that adaptively balances nuclear coverage and noise suppression. Our quantitative and qualitative analyses further elucidate the intrinsic behavior of prompt generation and verify its robust performance under diverse hyperparameter settings.

- We conduct extensive experiments on three challenging benchmarks, where SPROUT consistently achieves remarkable performance gains ($+8.2\%$ AJI on MoNuSeg). These highlight the potential of robust prompt generation and patch-based decomposition to unlock the zero-shot capabilities of vision foundation models in histopathology.

## 2 RELATED WORK

**Prompt Engineering for SAM.** SAM (Kirillov et al., 2023), as a vision foundation model, enables powerful zero-shot and class-agnostic segmentation via point, box, and coarse mask prompts. However, its performance in precise object delineation heavily depends on the accuracy and placement of input prompts. This has spurred a growing body of work on automatic prompt generation, as large-scale manual guidance is impractical in real-world clinical settings. Recent work includes embedding-oriented prompt representation learning (Luo et al., 2023; Yue et al., 2024; Li et al., 2025; Yan et al., 2025), detector-based prompt generation (Wu et al., 2023; Zheng et al., 2024; Xu et al., 2024; Xie et al., 2025), and heuristic-driven approaches (Gao et al., 2024). Meanwhile, prototype-guided methods (Zhang et al., 2024c; Wang et al., 2025) exploit feature correspondences between reference and target images to improve representation learning. Building on this idea, several training-free approaches generate prompts directly from such similarity mapping in natural images (Zhang et al., 2024a; Liu et al., 2024) and medical images (Liu et al., 2025). However, these methods are typically designed for scenarios with relatively few objects and rely on carefully curated reference images. By contrast, nuclear segmentation presents a far more challenging setting, involving thousands of densely packed and morphologically diverse objects within a single image. These limitations highlight the need for new strategies that can generate reliable prompts without external references while remaining computationally efficient.

**Nuclear Instance Segmentation.** Fully supervised nuclear instance segmentation approaches can be broadly grouped into three categories: contour-based (Chen et al., 2016), distance-mapping (Graham et al., 2019; He et al., 2021a), and detection-based (Jiang et al., 2023). While effective, these methods heavily rely on dense pixel-level annotations, which are costly and time-consuming. Semi- and weakly-supervised alternatives, particularly point-supervised approaches, have been explored as a new direction by transforming sparse labels into coarse pixel-level cues, such as Voronoi-based (Tian et al., 2020) or pseudo-edge maps (Yoo et al., 2019). But these approaches still suffer from unreliable pseudo masks and fail to adequately separate overlapping nuclei. With the advent of SAM, adapting foundation models to medical imaging has become a prominent direction. Methods such as MedSAM (Ma et al., 2024) and fine-tuning variants (Huang et al., 2024a;b) still require additional annotations and incur substantial computational overhead. To alleviate these constraints, other studies, including PromptNucSeg (Shui et al., 2024), UN-SAM (Chen et al., 2025b), and All-in-SAM (Cui et al., 2024), attempt to reduce these costs by training auxiliary prompters, introducing domain-adaptive feature tokens, or fine-tuning from self-generated masks. Building on these, we argue that competitive segmentation performance can be attained directly from SAM through proper prompt design and appropriate patch granularity without altering the model architecture.

## 3 SPROUT

The pipeline comprises three steps: (i) feature–prototype similarity mapping (Section 3.1), (ii) partial optimal transport scan with activation prompting (Section 3.2), and (iii) instance mask prediction with refinement (Section 3.3). As shown in Figure 2, SPROUT leverages stain priors to construct robust self-reference features, aligns them with prototypes through a theoretically grounded POT-Scan, and generates precise point prompts to guide SAM without additional training. The following subsections describe each stage in detail. The detailed theoretical analysis is provided in Appendix A.

### 3.1 FEATURE-PROTOTYPE SIMILARITY MAPPING

The first step is to extract self-reference features guided by stain priors and condense them into representative prototypes for subsequent matching. Given a pathology image $I \in \mathbb{R}^{H \times W \times 3}$, we partition it into $n$ overlapped patches of size $p \times p$ with stride $s$, denoted as $\{I^i\}_{i=1}^n$. Each patch is fed into the pretrained image encoder $f_\theta(\cdot)$ for feature extraction: $F^i = f_\theta(I^i)$. All patch-level features are then stitched together to reconstruct a global representation $F \in \mathbb{R}^{h \times w \times d}$, where $(h, w)$ is the spatial resolution of the encoded feature map and $d$ is the embedding dimension.

To derive self-reference masks, we first condition on stain color priors by transforming images into the optical density space: $OD = -\log(x/x_0)$, where $x \in \mathbb{R}^3$ is the observed RGB intensity and $x_0$ is the reference intensity. Using the normalized stain matrix $Q = [Q_H, Q_E]$, with the H&E absorbance profiles as columns, the stain concentration map $S = [S_H, S_E]^\top$ is obtained

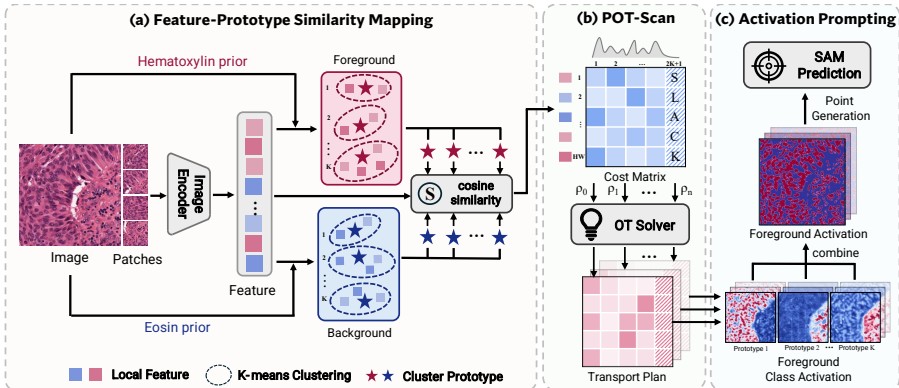

Figure 2: **SPROUT pipeline for point prompt generation.** It consists of three steps: (i) Feature–prototype similarity mapping: H&E stain priors is to identify high-confidence foreground and background regions, from which clustering extracts representative prototypes that serve as anchors for similarity matching; (ii) POT-Scan: a partial optimal transport scheme that progressively aligns features to prototypes, filtering ambiguous assignments through partial mass transport; (iii) Activation prompting: prototype-reweighted activations are aggregated into foreground maps, from which positive and negative point prompts are sampled to guide SAM-based instance prediction. For clarity, high-dimensional features are illustrated as squares and stars.

via linear decomposition: $S = Q^+ \cdot OD$, where $Q^+$ is the pseudoinverse. Otsu's thresholding (Appendix B.6) is then utilized to separate coarse foreground and background regions. Within each region, pixels with the top $t$ stain intensities are selected to construct high-confidence self-reference masks $M_{fg}$ and $M_{bg}$. To obtain compact feature prototypes and mitigate over-smoothing from dominant morphologies, we resize the self-reference masks to feature resolution and overlap them with the feature representation. Each class-specific feature set is then clustered into $K$ groups using $K$-means to derive representative prototypes:

$$\mathcal{P}_c = \{P_c^1, \ldots, P_c^K\} = \underset{\{P_c^1, \ldots, P_c^K\}}{\arg\min} \sum_{p \in \Omega} M_c(p) \min_{k=1,\ldots,K} ||F(p) - P_c^k||^2, \quad c \in \{fg, bg\}, \quad (1)$$

where $\Omega$ denotes the feature map spatial locations and $M_c \in \{0, 1\}$. The resulting prototypes $\{\mathcal{P}_{fg}, \mathcal{P}_{bg}\}$ are used as region-specific anchors for subsequent feature matching.

## 3.2 POT-SCAN AND ACTIVATION PROMPTING

**Preliminary.** To model feature-to-prototype alignment rigorously, we build on optimal transport (OT), which provides a principled framework for measuring distributional discrepancies. Optimal transport seeks the minimal cost of transporting one probability distribution onto another under marginal constraints. Given probability vectors $\mu \in \mathbb{R}^{n \times 1}$ and $\nu \in \mathbb{R}^{m \times 1}$ with cost matrix $C \in \mathbb{R}_+^{n \times m}$, the Kantorovich formulation (Kantorovich, 1942) solves:

$$\min_{T \in \mathbb{R}^{n \times m}} \langle T, C \rangle_F, \quad \text{s.t. } T\mathbf{1}_m = \mu, \ T^\top \mathbf{1}_n = \nu, \quad (2)$$

where $T$ is the transport plan and $\langle \cdot, \cdot \rangle_F$ denotes the Frobenius inner product. Relaxing the marginal constraints with divergences yields unbalanced OT. Adding entropic regularization (Cuturi, 2013) further enables efficient solutions via the Sinkhorn–Knopp algorithm (Knight, 2008). Further background, OT variants, and solver derivations are provided in Appendix A.1 and A.2.

**POT-Scan.** While OT offers a principled formulation, directly assigning features to prototypes in pathology is challenging due to noise and ambiguity. A naïve approach is to match each feature to its nearest prototype via cosine similarity:

$$C = 1 - \frac{\tilde{F} P^\top}{||\tilde{F}||_2 ||P||_2}, \quad (3)$$

where $P \in \mathbb{R}^{2K \times d}$ is the prototype matrix and $\tilde{F} \in \mathbb{R}^{hw \times d}$ is the flattened feature map. However, such point-wise matching is local and prone to collapse. Unbalanced OT relaxes marginal constraints

but still penalizes discarding noisy features. To address this, we adopt **partial OT**, which allows a fraction of the mass to remain unmatched and naturally filters ambiguous regions (Appendix A.3). Formally, transporting a fraction $\rho \in (0, 1]$ of the mass is posed as:

$$\min_{T \in \Pi} \ \langle T, C \rangle_F + \lambda KL(T^\top \mathbf{1}_N || \frac{\rho}{M} \mathbf{1}_M),$$

$$\text{s.t.} \ \Pi = \{T \in \mathbb{R}_+^{N \times M} | T\mathbf{1}_M \leq \frac{1}{N}\mathbf{1}_N, \ \mathbf{1}_M T^\top \mathbf{1}_N = \rho\} \tag{4}$$

where $N = h \times w$ denotes the number of source features, assumed to follow a uniform mass distribution since each feature is equally weighted, and $M = 2K$ is the number of target prototypes. This leads to our POT-Scan, where the transport ratio $\rho$ is progressively increased: starting from a small initial value $\rho_0$ that favors easy feature–prototype matches and gradually incorporating more ambiguous features until a stopping criterion is met.

Although conceptually intuitive, Eq.(4) cannot be solved directly using standard scaling algorithms due to non-normalized constraints. Following the reformulation of (Zhang et al., 2024b), we append a slack column to absorb the residual $1 - \rho$ mass, thereby restoring normalized marginals and enabling efficient Sinkhorn-based optimization. Detailed proofs, solver derivation, and corresponding pseudo-code are provided in Appendix A.3, A.4, and A.5.

**Activation Prompting.** Given optimal transport plan $T^\star$, features are reweighted as $F^\star = \tilde{F} \odot T^\star \in \mathbb{R}^{hw \times 2K}$. These are mapped back to the image space via the resizing operator $\mathcal{R}$ and refined with DenseCRF, producing activation maps $F' = \text{CRF}(\mathcal{R}(F^\star)) \in \mathbb{R}^{H \times W \times 2K}$. Foreground and background activations are aggregated as $[F'_{fg}, F'_{bg}] = [\sum_k F'^k_{fg}, \sum_k F'^k_{bg}]$, which are then binarized using Otsu's thresholding. Combining these with the initial high-confidence masks yields positive points through a watershed-based procedure (Appendix B.6), while negative points are uniformly sampled from expanded background masks with an additional stride to ensure sufficient nuclear coverage. The process terminates once multiple compact regions merge into a large connected component, as further expansion risks conflating distinct nuclei. This balances robust assignments with the gradual inclusion of difficult features, resulting in stable and informative prompts for SAM.

### 3.3 INSTANCE MASK PREDICTION AND REFINEMENT

The final step is to generate instance-level nuclear masks from activation-derived prompts and refine them to correct boundary errors from overlapping cells and weak edges. To capture fine-grained nuclear structures, we perform inference at the patch level, which allows SAM to better localize individual nuclei. Positive and negative prompts further help separate closely packed nuclei from surrounding tissue (Detailed illustrations are in Appendix E.1). When a nucleus has multiple positive cues, each positive together with $y$ nearest negatives is provided to SAM within its patch, and highly overlapped predictions are merged. Although weak inter-nuclear boundaries may lead to adjacent nuclei being predicted into a single instance, fragmentation is rare due to their homogeneous interiors. To this end, we introduce *containment-aware non-maximum suppression (NMS)* that penalizes large masks enclosing multiple smaller nuclei. Specifically, we apply a $\tanh$-based decay penalty proportional to the number of contained instances and combine SAM's confidence $S_{\text{SAM}}$ with the normalized hematoxylin-channel response $S'_H$ into a unified score $S = S_{\text{SAM}} + S'_H$ for filtering. This complementary strategy leverages both morphological consistency and biological priors, suppressing false positives and improving boundary delineation. Further implementation details of containment-aware NMS are provided in Appendix D.3.

## 4 EXPERIMENTS

We evaluate SPROUT on three benchmark datasets, MoNuSeg (Kumar et al., 2017), CPM17 (Vu et al., 2019), and TNBC (Naylor et al., 2018), using instance-level metrics (AJI, PQ, DQ, SQ) and the semantic-level Dice coefficient. Complete dataset statistics and metric definitions are provided in Appendix B.1 and B.2. Section 4.1 reports comparisons with state-of-the-art nuclear instance segmentation methods, and Section 4.2 (with Appendix D) presents ablations that validate the key components of SPROUT. We further analyze robustness through hyperparameter sensitivity studies (Section 4.3). Additional implementation details are included in Appendix B.3.

Table 1: **Performance evaluations of nuclear instance segmentation.** We benchmark methods across fully-, weakly-, self-supervised, and SAM-based approaches on MoNuSeg and CPM17 datasets. Segmentation performance is reported using AJI (↑), PQ (↑), DQ (↑), SQ (↑), and Dice (↑). Best results are highlighted in **bold**, and second-best are underlined.

| Method | SAM | Supervision | MoNuSeg | | | | | CPM17 | | | | |
|---|---|---|---|---|---|---|---|---|---|---|---|---|
| | | | AJI | PQ | DQ | SQ | Dice | AJI | PQ | DQ | SQ | Dice |
| U-Net_{MICCAI'15} | ✗ | fully | 0.421 | 0.403 | 0.571 | 0.705 | 0.635 | 0.554 | 0.527 | 0.718 | 0.734 | 0.741 |
| SPN+IEN_{ISBI'22} | ✗ | point | 0.521 | 0.436 | 0.661 | 0.660 | 0.677 | 0.540 | 0.485 | 0.695 | 0.699 | 0.701 |
| SC-Net_{MEDIA'23} | ✗ | point | 0.539 | 0.450 | 0.648 | 0.694 | 0.732 | 0.561 | 0.486 | 0.692 | 0.703 | 0.698 |
| SAM_{CVPR'23} | ✓ | point⋆ | 0.061 | 0.262 | 0.384 | 0.751 | 0.353 | 0.135 | 0.469 | 0.601 | 0.781 | 0.329 |
| DES-SAM_{MICCAI'24} | ✓ | box | 0.463 | 0.429 | 0.621 | 0.691 | 0.672 | 0.512 | 0.517 | 0.735 | 0.704 | 0.688 |
| MedSAM_{Nat. Commun.'24} | ✓ | box⋆ | 0.502 | 0.327 | 0.514 | **0.752** | 0.687 | 0.648 | 0.559 | 0.788 | 0.706 | 0.793 |
| UN-SAM_{MEDIA'25} | ✓ | self | 0.482 | 0.477 | 0.656 | 0.728 | 0.792 | 0.581 | 0.574 | 0.734 | **0.782** | 0.795 |
| Med-SA_{MEDIA'25} | ✓ | fully | 0.511 | 0.493 | 0.679 | 0.727 | 0.772 | 0.565 | 0.564 | 0.731 | 0.772 | 0.806 |
| **SPROUT** | ✓ | automatic⋆ | **0.621** | **0.601** | **0.817** | 0.736 | **0.795** | **0.662** | **0.616** | **0.796** | 0.774 | **0.821** |

⋆ indicates use of pretrained weights, and the supervision column specifies the input prompt type.

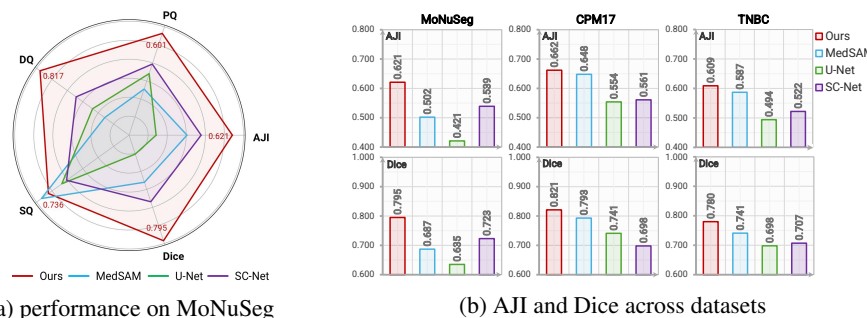

(a) performance on MoNuSeg         (b) AJI and Dice across datasets

Figure 3: **Performance comparison across supervision types.** SPROUT consistently outperforms SAM-based (MedSAM), fully supervised (U-Net), and point-supervised (SC-Net) models across datasets with superior effectiveness in segmentation.

## 4.1 MAIN RESULTS

**Comparison with State-of-the-Art Methods** We benchmark our method against other leading approaches in nuclear instance segmentation, including U-Net (Ronneberger et al., 2015), SPN+IEN (Liu et al., 2022), SC-Net (Lin et al., 2023), SAM (Kirillov et al., 2023), MedSAM (Ma et al., 2024), UN-SAM (Chen et al., 2025b), DES-SAM (Huang et al., 2024a), Med-SA (Wu et al., 2025). As summarized in Table 1, our method achieves the highest AJI and Dice scores and consistently outperforms all counterparts with up to 8.2% absolute gains in AJI on the challenging MoNuSeg dataset. Notably, the PQ scores of 0.601 on MoNuSeg and 0.616 on CPM17 demonstrate its strong ability to maintain object-level consistency. The baseline descriptions and reproduction details are provided in Appendix B.4. Comparative segmentation visualizations from natural-image segmentation models are provided in Appendix C.2 to highlight their limitations in pathology.

**Analysis by Supervision Type and Visualization.** Figure 3 shows our method surpasses MedSAM, U-Net, and SC-Net from different supervision regimes without annotations and training. We further make a visual comparison in Figure 4. SPROUT produces clean, non-overlapping masks in challenging cases with nuclei-tissue color similarity or light stain. By automatically generating smart positive–negative to constrain predictions, SPROUT fully exploits SAM's capacity and demonstrates strong generalizability and robustness across datasets and background appearances. Meanwhile, it offers an efficient alternative to supervised approaches, making training-free and annotation-free on pathological images possible while still achieving superior instance- and semantic-level accuracy. Additional qualitative examples across datasets are provided in Appendix C.1.

## 4.2 ABLATION STUDIES

We conduct ablation studies to evaluate the core components of SPROUT, focusing on two key questions: (i) How does SPROUT generate reliable point prompts for SAM prediction effectively? (ii) Which post-processing strategy ensures accurate instance mask selection?

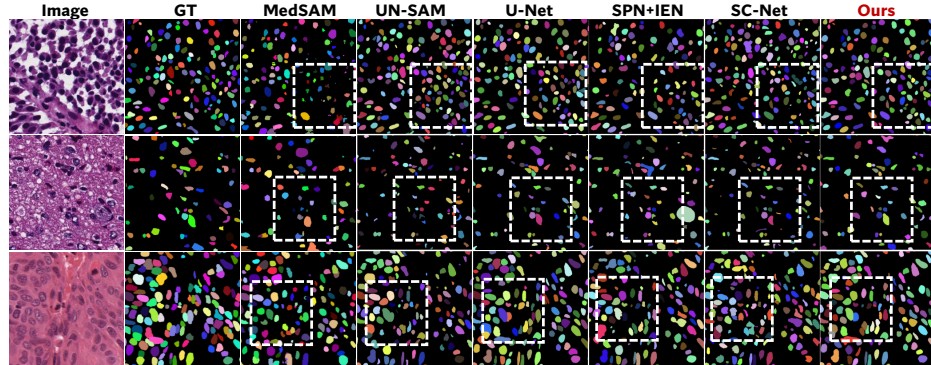

Figure 4: **Visualization of instance segmentation results from different methods.** SPROUT delivers more correct instances with fewer overlaps. The highlighted regions show distinct differences.

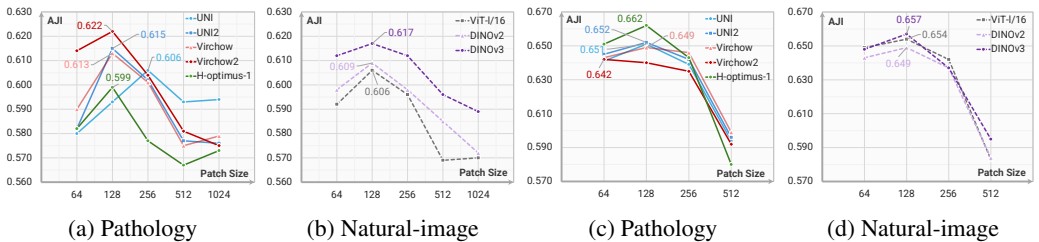

(a) Pathology      (b) Natural-image      (c) Pathology      (d) Natural-image

Figure 5: **Performance comparison of pathology- and natural-image-based backbones.** On MoNuSeg (a, b) and CPM17 (c, d), the self-reference mask strategy mitigates the domain gap and yields competitive performance, with the best AJI at patch size $128 \times 128$ matching nuclear scale.

**Feature Extractors.** To assess the effect of the proposed self-reference mask strategy, we evaluate feature extractors trained on both pathology (UNI, UNI2 (Chen et al., 2024), Virchow (Vorontsov et al., 2024), Virchow2 (Zimmermann et al., 2024), H-optimus-1 (Bioptimus, 2025)) and natural images (ViT-l/16 (Dosovitskiy et al., 2020), DINOv2 (Oquab et al., 2024), DINOv3 (Siméoni et al., 2025)) using the MoNuSeg and CPM17 datasets. The Appendix B.5 provides further details on each backbone. As shown in Figure 5, both types of backbones yield comparable AJI scores, typically within 1%. Virchow2 and DINOv3 achieve the best performance on MoNuSeg, while H-optimus-1 and DINOv3 perform best on CPM17. These findings validate that the proposed self-reference strategy incorporates image-specific H&E color priors to refine scales and allows cross-domain transfer without specialized fine-tuning. The consistent peaks of AJI at a patch size of $128 \times 128$ align with the feature extractor's receptive field relative to cell sizes. Overly large patches ($\geq 512$) dilute nuclear signals and overly small ones risk fragmenting them. Additional cross-dataset results demonstrating the robustness of self-reference are included in Appendix D.1.

**SAM Variants.** Since SPROUT relies on SAM for instance generation, we analyze how model size influences segmentation. Figure 6 reports AJI for large, base-plus, small, and tiny variants under different patch sizes. Splitting images into moderate patches improves AJI relative to whole-image input, while overly small patches provide little gain and increase computational cost by fragmenting context and amplifying noise. Large and base-plus models perform best, but the advantage of large over base-plus is minor since nuclei within each patch are relatively homogeneous. Smaller variants also remain competitive with patch inputs, suggesting practical value in resource-limited settings.

**Point Generation.** In Figure 7a, Otsu-based masks improve performance but remain unsatisfactory because color-only separation cannot resolve nuclei with close foreground–background intensities or noise. High-confidence masks improve robustness by filtering out unreliable regions. Feature extraction with similarity mapping alone provides limited benefit, since the quality of reference features depends on the initial masks and can still be affected by noise. Balanced OT offers marginal improvement as it enforces the assignment of ambiguous pixels to prototypes. In contrast, partial OT delivers the best performance by selectively transporting low-cost, high-confidence matches and scanning through the image to capture reliable regions for point generation. See Appendix D.2 for full quantitative results and further point quality analysis.

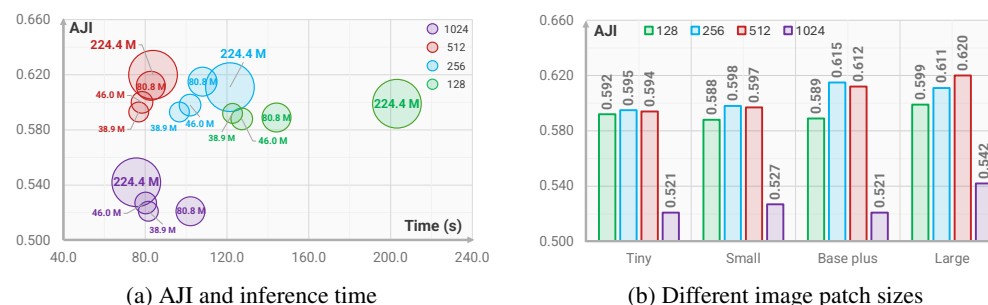

(a) AJI and inference time        (b) Different image patch sizes

Figure 6: **Relationship between SAM variants and patch size.** Appropriate patch sizes narrow the performance gap between large and small SAM variants and enable flexible deployment across resource settings. The best performance is obtained with the large SAM at patch size $512 \times 512$. Results are reported on the MoNuSeg dataset with a fixed patch overlap ratio of $0.5$.

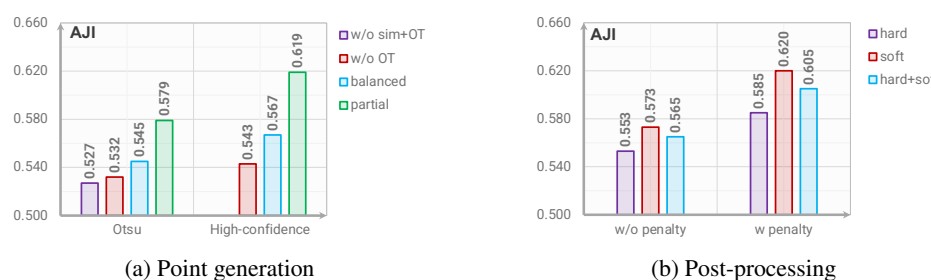

(a) Point generation        (b) Post-processing

Figure 7: (a) Sim: similarity mapping. High-confidence masks yield cleaner reference features with modest gains. Partial OT outperforms balanced OT by enabling more flexible assignments. (b) Adding a containment-aware penalty consistently improves AJI across NMS types. Soft NMS achieves the best performance by allowing mild overlap, which better matches practical morphology.

**Post-processing.** Figure 7b compares different NMS strategies with and without the containment-aware penalty. Without the penalty, large masks containing multiple nuclei are often retained, as their low IoU with smaller nucleus masks prevents them from being identified as redundant during NMS. Introducing the penalty improves AJI by about $5\%$, explicitly discouraging such scenarios. Among the selection strategies, soft NMS achieves the best balance by reducing redundant overlaps while still allowing partial overlap in dense regions, consistent with the biological reality of clustered nuclei. The hybrid strategy is less suitable because it enforces strict separation rather than acceptable overlaps. Ablations of soft NMS decay functions and score strategies are in Appendix D.3.

**Class Activation.** Figure 8 illustrates the prototype activation after POT-scan. Each prototype emphasizes distinct morphological patterns, and their combination recovers foreground structures closely aligned with ground truth. This confirms that $K$-means clustering of features produces discriminative prototypes, enabling robust capture of nuclei even under subtle feature diversity.

### 4.3 SENSITIVITY ANALYSIS

We analyze the impact of key hyperparameters on segmentation performance to better understand SPROUT's behavior under different configurations. Figure 9 reports results for point generation and mask prediction stages. Across all settings, SPROUT remains stable, with performance varying by less than $3\%$ except for extreme parameters. This robustness indicates that the framework requires little hyperparameter tuning in practice. Experimental details are provided in Appendix B.3.

**Point Generation.** In Figure 9a and 9b, the high-confidence mask ratio peaks around $0.6$ and decreases as the ratio approaches $1$. This indicates that a moderate ratio balances reliable regions with sufficient coverage, while extreme values either lack generalization or introduce noise. A similar trend is observed for the initial transported weight. The stride used in POT-Scan is relatively milder in Figure 9c. Small strides cause the process to stop early when nuclei are densely packed, while large strides may skip fine details. For the number of $K$-means clusters, small values fail to capture the diversity of foreground and background features. Performance improves as the cluster number increases and stabilizes once $K \geq 3$, as shown in Figure 9d.

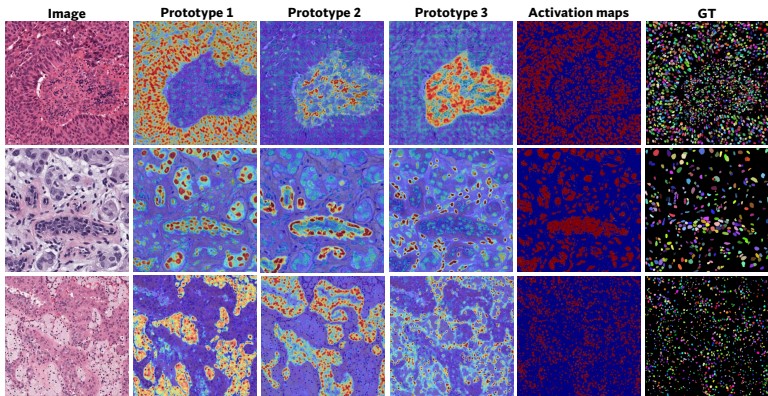

Figure 8: **Visualization of class activation on MoNuSeg dataset.** Different prototypes emphasize complementary tissue morphologies. Their aggregation activation maps align with the ground-truth. Clustering-based prototypes capture diverse feature variations for accurate localization.

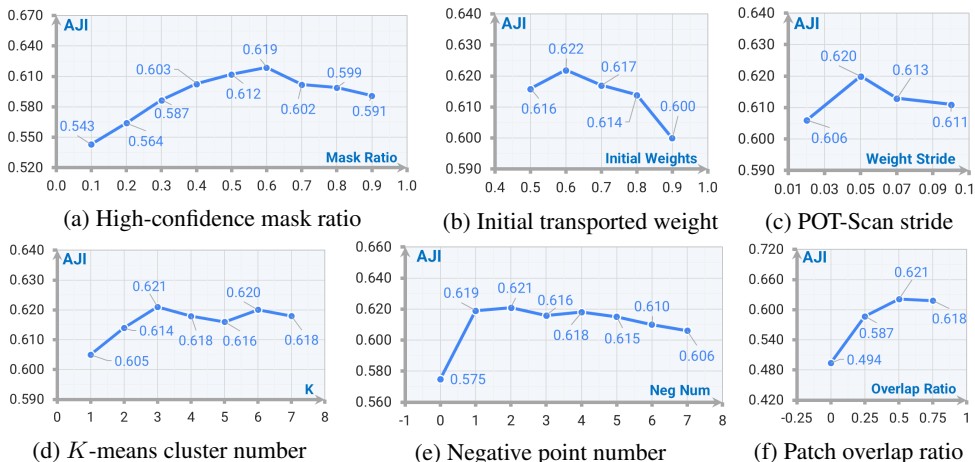

(a) High-confidence mask ratio  (b) Initial transported weight  (c) POT-Scan stride

(d) $K$-means cluster number  (e) Negative point number  (f) Patch overlap ratio

Figure 9: **Hyperparameter Sensitivity Analysis.** We evaluate AJI on the MoNuSeg dataset across six representative parameters. The results reveal only minor variation (less than 3%) under wide changes, confirming that SPROUT is robust to hyperparameter choices and requires minimal tuning.

**Mask Prediction.** In Figure 9e, introducing a small number of negative points improves accuracy by excluding ambiguous regions that a single positive point cannot separate, as illustrated in Appendix E.1. However, too many negative prompts make the model conservative, leading to smaller and fragmented predictions. The prediction patch overlap ratio has the strongest impact in this stage, as in Figure 9f. Without overlap, AJI is low due to border artifacts. Moderate overlap around 0.5 is critical for handling boundary regions, while avoiding redundant computation.

## 5 CONCLUSION

We presented **SPROUT**, a fully training-free framework for nuclear instance segmentation that generates prompts automatically without annotations. SPROUT introduces a self-reference strategy and a theoretically-grounded POT-Scan scheme to achieve precise feature representation and reduce domain gaps. By guiding SAM with automatically generated point prompts and applying a containment-aware NMS for lightweight refinement, SPROUT yields accurate and efficient segmentation without needs for fine-tuning or adapter modules. Extensive experiments show that SPROUT outperforms previous state-of-the-art methods across multiple datasets and provides new insights into the behavior of prompting-based pipelines. Beyond nuclear segmentation, this work points toward broader potential of bridging domain gaps via cross-domain priors, offering a path to more robust and adaptable medical imaging models. We acknowledge current limitations, including reliance on SAM for boundary precision and restriction to H&E images, but view SPROUT as a promising, scalable, and trustworthy step toward reliable, end-to-end AI integration in digital pathology.

ETHICS STATEMENT

All nuclear datasets used in our experiments (MoNuSeg, CPM17, and TNBC) are publicly available and employed in accordance with their respective licenses. No additional human or animal subjects were involved in this study. This method involves medical data and should be properly regulated to avoid compromising privacy and security. Importantly, this method is intended for research purposes only and must not be employed for clinical decision-making. The authors declare no competing interests.

REPRODUCIBILITY STATEMENT

The paper provides all necessary information to reproduce the main experimental results, including experimental settings and compute resources in Section 3 and Appendix B. The datasets used are publicly available. All assumptions and complete proof of the partial optimal transport formulation and solver are presented in Appendix A.2–A.5.

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

## APPENDIX

## A  THEORETICAL ANALYSIS

In this section, we provide the theoretical foundations and complete proofs for Section 3.2. We begin with a brief discussion of applications of optimal transport (OT) to set the stage, followed by a review of its standard formulation and the entropic scaling algorithm for efficient computation. We then derive the partial OT formulation adopted in the POT-Scan module by extending from unbalanced OT. In particular, we show that introducing a slack column transforms partial OT into an equivalent unbalanced OT problem, which can be efficiently solved using Sinkhorn-based methods. Finally, we present the algorithm employed to solve partial OT.

### A.1  BACKGROUND

Optimal Transport (OT) (Villani et al., 2008) provides a mathematical framework for aligning one probability measure with another by finding the most cost-efficient way to reallocate mass. Classical OT corresponds to the case where exact mass preservation is enforced. Variants such as unbalanced OT relax the constraint to handle discrepancies in total mass. Adding entropic regularization (Zhang et al., 2023; Cuturi, 2013) to the objective function enables efficient approximation via Sinkhorn iterations, making large-scale applications practical. Beyond its theoretical elegance, OT has become a versatile tool in modern machine learning. It has supported advances in generative modeling by providing stable training objectives through Wasserstein distances (Gulrajani et al., 2017), in semi-supervised learning by enabling label propagation as a transport problem (Tai et al., 2021), and in domain adaptation by aligning feature distributions across domains (Courty et al., 2016).

## A.2 SCALING ALGORITHM FOR OPTIMAL TRANSPORT

The optimal transport problem can be formulated as a minimization task over transport plans. Given probability vectors $\mu \in \mathbb{R}^{N \times 1}$, $\nu \in \mathbb{R}^{M \times 1}$, along with a cost matrix $C \in \mathbb{R}_+^{N \times M}$ defined on joint space, the objective function is written as:

$$\min_{T \in \mathbb{R}^{N \times M}} \langle T, C \rangle_F + \phi(T \mathbf{1}_M, \mu) + \psi(T^\top \mathbf{1}_N, \nu) \tag{5}$$

where $T \in \mathbb{R}^{N \times M}$ denotes the transportation plan, $\langle \cdot, \cdot \rangle_F$ is the Frobenius product. $\phi$ and $\psi$ are convex marginal distribution constraints, $\mathbf{1}_M \in \mathbb{R}^{M \times 1}$, $\mathbf{1}_N \in \mathbb{R}^{N \times 1}$ are all one vectors. This is the classical Kantorovich formulation (Kantorovich, 1942) if $\phi$ and $\psi$ are equality constraints. By relaxing the marginal constraints via KL divergence or inequality penalties, the problem generalizes to the unbalanced OT as described in Section A.3.

To make this problem computationally tractable, Cuturi (Cuturi, 2013) proposed entropic regularization. Adding the entropy term $-\epsilon \mathcal{H}(T)$ to objective function leads to the following formulation:

$$\begin{aligned}
\langle T, C \rangle_F - \epsilon \mathcal{H}(T) &= \epsilon \langle T, C/\epsilon + \log T \rangle_F \\
&= \epsilon \langle T, \log \frac{T}{\exp(-C/\epsilon)} \rangle_F \\
&= \epsilon KL(T || \exp(-C/\epsilon)),
\end{aligned} \tag{6}$$

Furthermore, Eq.(6) can be reformulated as:

$$\min_{T \in \mathbb{R}_+^{N \times M}} \epsilon KL(T || \exp(-C/\epsilon)) + \phi(T \mathbf{1}_M, \mu) + \psi(T^\top \mathbf{1}_N, \nu) \tag{7}$$

Define the proximal operator as:

$$\text{prox}_{f/\epsilon}^{KL}(y; z) = \arg\min_{x \geq 0} f(x, z) + \epsilon KL(x || y), \tag{8}$$

where $z$ is the fixed parameter of the function $f$. In our case, $z$ corresponds to the marginal distributions while $f$ represents the associated marginal constraints $\phi$ or $\psi$. Then Eq.(7) can be solved approximately using Alg.(1).

---

**Algorithm 1** Generalized scaling algorithm

---

1: **Input:** Cost $C$, regularization $\epsilon > 0$, marginals $\mu \in \mathbb{R}_+^N$, $\nu \in \mathbb{R}_+^M$
2: $Q \leftarrow \exp(-C/\epsilon)$     ▷    $Q_{ij} = e^{-C_{ij}/\epsilon}$
3: $b \leftarrow \mathbf{1}_n$
4: **while** not converged **do**
5:     $x \leftarrow Qb$
6:     $\tilde{a} \leftarrow \text{prox}_{\phi/\epsilon}^{KL}(x; \mu)$
7:     $a \leftarrow \tilde{a} \oslash x$     ▷    elementwise division
8:     $y \leftarrow Q^\top a$
9:     $\tilde{b} \leftarrow \text{prox}_{\psi/\epsilon}^{KL}(y; \nu)$
10:     $b \leftarrow \tilde{b} \oslash y$
11: **end while**
12: **return** $T^\star = \text{diag}(a) \, Q \, \text{diag}(b)$

---

These updates can be interpreted as Bregman projections with respect to the KL divergence onto convex sets defined by the marginal constraints (Benamou et al., 2015). Alternating such projections is guaranteed to converge, and the diagonal scaling form makes each iteration linear in the number of nonzero entries of $Q$. The entropic regularization enforces strict positivity, prevents sparsity and collapse of the transport plan, and enhances numerical stability. Intuitively, the scaling vectors $a, b$ can be viewed as per-row and per-column adjustment factors, respectively. Multiplying by $a$ rescales entire rows to match $\mu$, while multiplying by $b$ rescales columns to align with $\nu$. The iteratively alternating drives the transport plan $T$ to satisfy the marginal structure.

As a result, whenever an optimal transport problem can be reformulated with suitable marginal constraints into the form of Eq.(5), the corresponding proximal operators can be derived as in Eq.(8). This allows the problem to be efficiently solved using Alg.(1).

### A.3 DERIVATION FROM STANDARD OT TO PARTIAL OT

When strict equality constraints are not enforced, one may allow mass to be created or discarded. This leads to the unbalanced OT formulation where deviations from the marginals are penalized by a $KL$ divergence. Assuming a uniform source distribution, Eq. (5) can be expressed as:

$$\min_{T \in \Pi} \ \langle T, C \rangle_F + \lambda KL(T^\top \mathbf{1}_N || \frac{1}{M} \mathbf{1}_M)$$
$$\text{s.t. } \Pi = \{T \in \mathbb{R}_+^{N \times M} \,|\, T\mathbf{1}_M = \frac{1}{N} \mathbf{1}_N\}, \tag{9}$$

where $\lambda$ is the regularization weight factor. Here, the row sums are fixed to the uniform source distribution, while the column sums are softly penalized toward uniformity.

Although unbalanced OT relaxes the marginal constraints, it still penalizes discrepancies between the transported and target mass. As a result, even ambiguous or noisy features are still encouraged to be moved, potentially degrading the quality of the solution. To address this limitation, we adopt the partial OT formulation, which explicitly controls the amount of total transported mass. Instead of hard-thresholding unreliable features, partial OT allows the model to reweigh and selectively transport a subset of the source samples by solving:

$$\min_{T \in \Pi} \ \langle T, C \rangle_F + \lambda KL(T^\top \mathbf{1}_N || \frac{\rho}{M} \mathbf{1}_M)$$
$$\text{s.t. } \Pi = \{T \in \mathbb{R}_+^{N \times M} | T\mathbf{1}_M \le \frac{1}{N} \mathbf{1}_N, \ \mathbf{1}_N^\top T\mathbf{1}_M = \rho\}, \tag{10}$$

where $N = h \times w$ is the uniform source feature and $M = 2K$ is the number of target prototypes as described in Eq.(4). $\rho$ specifies the total transported mass and will increase gradually. Intuitively, partial OT still respects the distributional structure but enables progressive selection of reliable samples. Low-cost correspondences are favored first, while noisier or ambiguous features can be safely ignored or deferred until $\rho$ increases. This mechanism provides a principled way to suppress noise while guiding the optimization toward a globally consistent transport plan.

Mathematically, we follow prior work (Caffarelli & McCann, 2010; Chapel et al., 2020; Zhang et al., 2024b) to reformulate the partial OT problem as an unbalanced OT problem that can be solved efficiently with scaling algorithms. The key idea is to introduce a slack column into the marginal distribution to absorb the unselected mass $1 - \rho$, thereby turning the global mass constraint into a marginal one. Specifically, the slack column is denoted as $\eta \in \mathbb{R}^{N \times 1}$ to absorb the remaining mass and form the extended coupling:

$$\hat{T} = [T, \eta] \in \mathbb{R}^{N \times (M+1)}, \quad \hat{C} = [C, \mathbf{0}_N].$$

Imposing row-sum equality to the uniform source and total-mass accounting, we get:

$$\hat{T}\mathbf{1}_{M+1} = \frac{1}{N} \mathbf{1}_N, \quad \mathbf{1}_N^\top \eta = 1 - \rho, \quad \mathbf{1}_N^\top T\mathbf{1}_M = \rho,$$

Thus,

$$\hat{T}^\top \mathbf{1}_N = \left[ \begin{array}{c} T^\top \mathbf{1}_N \\ \eta^\top \mathbf{1}_N \end{array} \right] = \left[ \begin{array}{c} T^\top \mathbf{1}_N \\ 1 - \rho \end{array} \right]. \tag{11}$$

Let the target column-mass prior be:

$$\beta = \left[ \begin{array}{c} \frac{\rho}{M} \mathbf{1}_M \\ 1 - \rho \end{array} \right],$$

we can get the KL-penalized unbalanced surrogate of partial OT as follows:

$$\min_{\hat{T} \in \Phi} \langle \hat{T}, \hat{C} \rangle_F + \lambda KL(\hat{T}^\top \mathbf{1}_N || \boldsymbol{\beta})$$
$$\text{s.t. } \Phi = \{\hat{T} \in \mathbb{R}_+^{N \times (M+1)} | \hat{T}\mathbf{1}_{M+1} = \frac{1}{N} \mathbf{1}_N\}. \tag{12}$$

However, the KL term is soft. Eq.(12) does not guarantee the mass of the last column to be strictly $1 - \rho$. To recover the exact partial-OT constraint, a weighted $KL$ constraint is employed to control the constraint strength for each class:

$$\hat{KL}(\hat{T}^\top \mathbf{1}_N || \beta; \hat{\lambda}) = \sum_{i=1}^{M+1} \lambda_i [\hat{T}^\top \mathbf{1}_N]_i \log \frac{[\hat{T}^\top \mathbf{1}_N]_i}{\beta_i}, \tag{13}$$

with

$$\hat{\lambda} = \left[ \begin{array}{c} \lambda \mathbf{1}_M \\ +\infty \end{array} \right].$$

This yields the final equivalent formulation:

$$\min_{\hat{T} \in \Phi} \ \langle \hat{T}, \hat{C} \rangle_F + \hat{K}L(\hat{T}^\top \mathbf{1}_N || \beta; \hat{\lambda}).$$

$$\text{s.t. } \Phi = \{ \hat{T} \in \mathbb{R}_+^{N \times (M+1)} | \hat{T} \mathbf{1}_{M+1} = \frac{1}{N} \mathbf{1}_N \}$$

(14)

The weighted KL makes the slack mass non-negotiable while keeping the real columns softly regularized. So low-cost correspondences are selected first, and ambiguous features can be safely left in the slack. The extended optimal plan is consistent with the original one, and the first $M$ columns of the extended solution align with the optimal plan of the partial OT problem. The proof is provided in the next section.

### A.4 PROOF OF EQUIVALENCE WITH PARTIAL OT

In this section, we present the full proof that $\tilde{T}^\star$, which is the first M columns of the extended optimal transport plan $\hat{T}^\star$, corresponds exactly to the optimal plan $T^\star$ of the partial OT problem.

*Proof.* Assume the optimal extended plan is:

$$\hat{T}^\star = [\tilde{T}^\star, \eta^\star] \in \mathbb{R}^{N \times (M+1)}, \tilde{T}^\star \in \mathbb{R}^{N \times M}.$$

The weighted KL penalty expands as:

$$\hat{K}L(\hat{T}^{\star\top} \mathbf{1}_N || \beta; \hat{\lambda}) = \sum_{i=1}^{M} \lambda_i [\tilde{T}^{\star\top} \mathbf{1}_N]_i \log \frac{[\tilde{T}^{\star\top} \mathbf{1}_N]_i}{\beta_i} + \lambda_{M+1} \eta^{\star\top} \mathbf{1}_N \log \frac{\eta^{\star\top} \mathbf{1}_N}{1 - \rho}$$

$$= \lambda KL(\tilde{T}^{\star\top} \mathbf{1}_N || \frac{\rho}{M} \mathbf{1}_M) + \lambda_{M+1} \eta^{\star\top} \mathbf{1}_N \log \frac{\eta^{\star\top} \mathbf{1}_N}{1 - \rho}.$$

(15)

Taking the limit $\lambda_{M+1} \to +\infty$ forces the slack column to satisfy $\eta^{\star\top} \mathbf{1}_N = 1 - \rho$, otherwise the objective would diverge.

By construction, the extended plan satisfies the row constraint

$$\hat{T}^\star \mathbf{1}_{M+1} = \frac{1}{N} \mathbf{1}_N.$$

This can be written as

$$\tilde{T}^\star \mathbf{1}_M + \eta^\star = \frac{1}{N} \mathbf{1}_N, \eta^\star > 0,$$

we obtain

$$\tilde{T}^\star \mathbf{1}_M \le \frac{1}{N} \mathbf{1}_N.$$

In addition, the total transported mass of the first $M$ column is

$$\mathbf{1}_N^\top \tilde{T}^\star \mathbf{1}_M = \mathbf{1}_N^\top \hat{T}^\star \mathbf{1}_M - \mathbf{1}_N^\top \eta^\star = 1 - (1 - \rho) = \rho.$$

Therefore,

$$\tilde{T}^\star \in \{ \tilde{T}^\star \in \mathbb{R}^{N \times M} | \tilde{T}^\star \mathbf{1}_M \le \frac{1}{N} \mathbf{1}_N, \mathbf{1}_N^\top \tilde{T}^\star \mathbf{1}_M = \rho \},$$

which is precisely the feasible set of the partial OT problem.

Lastly, the cost of the extended problem is

$$\langle \hat{T}^\star, \hat{C} \rangle_F + \hat{K}L(\hat{T}^{\star\top} \mathbf{1}_N || \boldsymbol{\beta}; \hat{\lambda}) = \langle [\tilde{T}^\star, \eta^\star], [C, \mathbf{0}_n] \rangle_F + \lambda KL(\tilde{T}^{\star\top} \mathbf{1}_N, \frac{\rho}{M} \mathbf{1}_M)$$

$$= \langle \tilde{T}^\star, C \rangle_F + \lambda KL(\tilde{T}^{\star\top} \mathbf{1}_N, \frac{\rho}{M} \mathbf{1}_M)$$

(16)

This is exactly the objective of the partial OT problem as in Eq.(10) evaluated at $\tilde{T}^{\star}$.

If $\tilde{T}^{\star}$ achieves a lower cost than $T^{\star}$ for the initial partial OT formula, it contradicts the optimality of $T^{\star}$.

If $T^{\star}$ had strictly lower cost for Eq.(16), then $\tilde{T}^{\star}$ would no longer achieve the optimum, which would contradict the optimality of $\hat{T}^{\star}$.

As a result, by convexity of the objective, $\tilde{T}^{\star} = T^{\star}$. Dropping the last column of $\hat{T}^{\star}$, we achieve the optimal transport plan for the partial OT problem. $\qquad\square$

## A.5 SOLVER FOR PARTIAL OT

Adding an entropy regularization term $-\epsilon\mathcal{H}(\hat{T})$ to Eq.(14) also enables the efficient scaling algorithm. We denote:

$$Q = \exp(-C/\epsilon), f = \frac{\lambda}{\lambda + \epsilon}, \alpha = \frac{1}{N}\mathbf{1}_N$$

The optimal plan admits the standard scaling form:

$$\hat{T}^{\star} = \mathrm{diag}(a)\,Q\,\mathrm{diag}(b).$$

*Proof.* As in Section A.2, the main step is to compute the proximal operators corresponding to the constraints $\phi$ and $\psi$. To this end, let us first restate the Eq.(14) in a more general form:

$$\min_{T \in \Phi} \epsilon KL(T \,\|\, \exp(-C/\epsilon)) + \hat{K}L(T^{\top}\mathbf{1}_N \|\beta; \lambda),$$

$$\text{s.t.} \quad \Phi = \{T \in \mathbb{R}_+^{N \times M} \mid T\mathbf{1}_M = \alpha\} \tag{17}$$

where $C$ is the cost matrix, $\alpha$ is the source marginal.

The equality constraint $T\mathbf{1}_M = \alpha$ can be expressed as the indicator:

$$\phi(x; \alpha) = \begin{cases} 0, & x = \alpha, \\ +\infty, & \text{otherwise.} \end{cases}$$

Plugging this into the proximal operator directly gives: $\mathrm{prox}_{\phi/\epsilon}^{KL}(y; \alpha) = \alpha$.

For the weighted KL penalty, the proximal operator is defined as:

$$\mathrm{prox}_{\psi/\epsilon}^{KL}(y; \beta) = \arg\min_{x \geq 0} \hat{K}L(x\|\beta; \lambda) + \epsilon\,KL(x\|y) \tag{18}$$

$$= \arg\min_{x \geq 0} \sum_{i=1}^{M+1} \lambda_i\left(x_i \log \frac{x_i}{\beta_i} - x_i + \beta\right) + \epsilon\left(x_i \log \frac{x_i}{y_i} - x_i + y_i\right). \tag{19}$$

After dropping constants independent of $x$ and regrouping terms, we obtain:

$$\mathrm{prox}_{\psi/\epsilon}^{KL}(y; \beta) = \arg\min_{x \geq 0} \sum_{i=1}^{M+1} (\lambda_i + \epsilon)x_i \log x_i - \left(\lambda_i \log \beta_i + \lambda_i + \epsilon \log y_i + \epsilon\right)x_i.$$

Consider the generic function $g(x) = ax \log x - bx$ with $a > 0$,

its derivative is $g'(x) = a(1 + \log x) - b$, hence the minimizer is $x^{\star} = \exp(\frac{b-a}{a})$. Applying this result gives:

$$x_i^{\star} = \exp\left(\frac{\lambda_i \log \beta_i + \epsilon \log y_i}{\lambda_i + \epsilon}\right)$$

$$= \beta_i^{\frac{\lambda_i}{\lambda_i + \epsilon}}\, y_i^{\frac{\epsilon}{\lambda_i + \epsilon}}.$$

In vector notation, we write:

$$x^{\star} = \beta^{\circ f}\, y^{\circ(1-f)}, \quad f = \frac{\lambda}{\lambda + \epsilon},$$

where ∘ denotes the element-wise power.

Now, substituting the two proximal operators into the general scaling algorithm yields the updates:

$$a \leftarrow \frac{\alpha}{Qb}, \quad b \leftarrow \left(\frac{\beta}{Q^\top a}\right)^{\circ f},$$

where $Q = \exp(-C/\epsilon)$.

□

The pseudo-code of the scaling algorithm for partial OT is provided in Alg.(2)

---

**Algorithm 2** Scaling algorithm for partial OT

---

1: Initialize: Cost matrix $C$, $\epsilon$, $\lambda$, $\rho$, $N$, $K$, a large value $\iota$
2: $C \leftarrow [C, \mathbf{0}_N]$,
3: $\lambda \leftarrow [\lambda, ..., \lambda, \iota]^\top$
4: $\beta \leftarrow [\frac{\rho}{M}\mathbf{1}_M^\top, 1 - \rho]^\top$
5: $\alpha \leftarrow \frac{1}{N}\mathbf{1}_N$
6: $b \leftarrow \mathbf{1}_{M+1}$
7: $Q \leftarrow \exp(-C/\epsilon)$
8: $f \leftarrow \frac{\lambda}{\lambda+\epsilon}$
9: **while** $b$ does not converge **do**
10:     $a \leftarrow \frac{\alpha}{Mb}$
11:     $b \leftarrow (\frac{\beta}{M^\top a})^{\circ f}$,
12:     $T \leftarrow \mathrm{diag}(a)Q\mathrm{diag}(b)$
13: **end while**
14: **return** $T[:, :K]$

---

## B  ADDITIONAL IMPLEMENTATION DETAILS

### B.1  DATASETS

We conduct experiments on three challenging benchmark datasets of H&E stained histopathology images: MoNuSeg (Kumar et al., 2017), CPM17 (Vu et al., 2019), and TNBC (Naylor et al., 2018).

**(I) MoNuSeg.** MoNuSeg is a multi-organ nuclei segmentation dataset created from H&E-stained tissue images at $40\times$ magnification from the TCGA archive (Weinstein et al., 2013), containing 51 images at $1000\times1000$ resolution from 7 organs with a total of 30,837 individually annotated nuclear boundaries. The MoNuSeg dataset contains 37 training images and 14 testing images.

**(I) CPM17.** CPM17 contains 64 H&E stained histopathology images at $500 \times 500$ resolution with 7,670 annotated nuclei. Each image was scanned at $40\times$ magnification. CPM17 includes 32 images each for training and testing.

**(I) TNBC.** TNBC consists of 50 H&E stained histopathology images of size $500 \times 500$ from Triple Negative Breast Cancer (TNBC) patients, containing 4,028 nuclear annotations.

To facilitate patch-based processing for both feature extraction and SAM prediction, we use Lanczos interpolation (Lanczos, 1964) over $8 \times 8$ neighborhood to resize the MoNuSeg images to $1024 \times 1024$, and the CPM17 and TNBC images to $512 \times 512$.

### B.2  METRICS

Nuclear instance segmentation performance is evaluated using four instance-level metrics, including Detection Quality (DQ), Segmentation Quality (SQ), Panoptic Quality (PQ), and the Aggregated Jaccard Index (AJI), along with one semantic-level metric, the Dice coefficient (Dice). The detailed definitions are as follows:

Let $\mathcal{G} = \{G_i\}_{i=1}^{N_G}$ and $\mathcal{P} = \{P_j\}_{j=1}^{N_P}$ denote the sets of ground-truth and predicted instances, respectively. Define the IoU (Intersection over Union) as below:

$$\text{IoU}_{ij} = \frac{|G_i \cap P_j|}{|G_i \cup P_j|}.$$

**Dice.** Dice score measures overall pixel-level agreement and is insensitive to instance identities. It can be calculated using foreground overlap after binarization:

$$\text{Dice} = \frac{2\,|(\cup_i G_i) \bigcap (\cup_j P_j)|}{|\cup_i G_i| + |\cup_j P_j|}.$$

**AJI (Aggregated Jaccard Index).** AJI is an instance-aware Jaccard that penalizes both split and merged instances since the unmatched regions go to the denominator.

Let $\mathcal{M} \subseteq \{1, \ldots, N_G\} \times \{1, \ldots, N_P\}$ be a one-to-one matching between instances with $\text{IoU}_{ij} > 0$. Then AJI can be calculated as:

$$\text{AJI} = \frac{\sum_{(i,j)\in\mathcal{M}} |G_i \cap P_j|}{\sum_{(i,j)\in\mathcal{M}} |G_i \cup P_j| + \sum_{i:\,(i,\cdot)\notin\mathcal{M}} |G_i| + \sum_{j:\,(\cdot,j)\notin\mathcal{M}} |P_j|}.$$

**PQ, DQ, and SQ.** Detection Quality (DQ) quantifies the accuracy of instance detection by accounting for false positives and false negatives, while Segmentation Quality (SQ) measures the delineation accuracy of correctly matched instances. Panoptic Quality (PQ), defined as the product of DQ and SQ, provides a comprehensive metric that jointly reflects both detection and segmentation performance.

The matched pairs (true positive) are defined with a fixed threshold $\tau = 0.5$.

$$\text{TP} = \big\{(i,j) : \text{IoU}_{ij} > \tau\big\}.$$

Let $\text{FP} = N_P - |\text{TP}|$ and $\text{FN} = N_G - |\text{TP}|$. Then

$$\text{DQ} = \frac{|\text{TP}|}{|\text{TP}| + \frac{1}{2}\text{FP} + \frac{1}{2}\text{FN}},$$

$$\text{SQ} = \frac{1}{|\text{TP}|} \sum_{(i,j)\in\text{TP}} \text{IoU}_{ij},$$

$$\text{PQ} = \text{DQ} \times \text{SQ}.$$

In practice, the one-to-one correspondences between ground truth and predicted instances are established using a greedy matching strategy. The evaluation metrics are subsequently obtained by averaging results across all images.

### B.3 IMPLEMENTATION

**General Settings.** Our method is training-free and therefore does not require dataset partitioning. To ensure a fair comparison with the trained networks, we conduct evaluations on the fixed test sets of MoNuSeg and CPM17. For TNBC, we randomly split the data with an 80/20 ratio for training and testing. All experiments are performed on a single NVIDIA GeForce RTX 4090 GPU with 24 GB of memory. Each experiment is repeated three times, and the average performance is reported.

**Baseline Reproduction.** For baseline model comparison, we use the official source code released by the authors on GitHub and follow the recommended configurations in the original publications. U-Net predictions are converted into instance segmentation masks using the watershed algorithm. For baselines that require annotations, prompts are generated from the ground truth when no automatic prompt generation mechanism is available. Note that we employ automatic prompting rather than ground-truth point prompts for the vanilla SAM model. All reported numbers correspond to our reproduced results.

**Sensitivity Studies.** Sensitivity studies are conducted on the MoNuSeg dataset using a patch size of $128 \times 128$ and a stride of $64$ for feature extraction with Virchow2. Default settings are as follows: the initial high-confidence mask ratio is $0.6$ (varied in Figure 9a), the number of prototype centers is $3$ (changed in Figure 9d), the initial OT weight is $0.6$ (investigated in Figure 9b), and the POT-Scan weight stride is $0.05$ (analyzed in Figure 9c). The large SAM weights are employed, and images are tiled into $512 \times 512$ patches with an overlap ratio of $0.5$ (tuned in Figure 9f). For prediction, each positive point and two negative points are provided to SAM (altered in Figure 9e). Containment-aware soft NMS is applied in the post-processing stage.

### B.4 BASELINES

We provide detailed descriptions of the baseline models used for comparison. Their implementation settings and necessary adaptations are provided in Section B.3.

**U-Net.** U-Net (Ronneberger et al., 2015) is a classic convolutional neural network architecture for biomedical image segmentation. Its encoder–decoder design combines a contracting path for contextual extraction with an expansive path for precise localization through symmetric skip connections. This structure allows U-Net to learn fine-grained spatial details from limited training data, making it a widely adopted baseline for medical segmentation tasks.

**SPN+IEN.** SPN+IEN (Liu et al., 2022) is a weakly supervised framework that uses point annotations for nuclei segmentation. It separates the task into two complementary modules: a Semantic Proposal Network (SPN) that generates coarse foreground–background masks, and an Instance Encoding Network (IEN) that learns instance-aware pixel embeddings to distinguish neighboring nuclei. Such a design reduces annotation costs while maintaining strong segmentation performance.

**SC-Net.** Shape-Constrained Network (SC-Net) (Lin et al., 2023) integrates morphological shape priors into the learning process for nuclei instance segmentation. It employs a detection branch to localize nuclei and a segmentation branch to refine masks. Shape constraints are enforced to better handle overlapping or irregularly shaped nuclei.

**Segment Anything Model.** SAM (Kirillov et al., 2023) is a foundation segmentation model designed for interactive instance segmentation. It consists of an image encoder, a prompt encoder, and a lightweight mask decoder. By leveraging spatial or textual prompts (points, boxes, masks), SAM produces high-quality predictions in a zero-shot manner across diverse domains.

**DES-SAM.** DES-SAM (Huang et al., 2024a) is a distillation-enhanced adaptation of SAM for box-supervised nucleus segmentation. It incorporates a lightweight detection module to generate bounding-box prompts and uses a self-distillation prompting strategy to leverage SAM's pretrained knowledge while fine-tuning a small number of parameters. DES-SAM introduces an edge-aware loss to refine boundary quality as well. Together, these components allow DES-SAM to achieve accurate segmentation with limited supervision while preserving SAM's strong generalization ability.

**MedSAM.** MedSAM (Ma et al., 2024) adapts the SAM to the medical domain and serves as a foundation model for universal medical image segmentation. It is trained on an unprecedented dataset of over 1.5 million image–mask pairs, spanning 10 imaging modalities and over 30 cancer types. In this way, it captures a wide spectrum of anatomical structures and pathological conditions. Like SAM, MedSAM is also a promptable segmentation system, where the user inputs bounding boxes to guide the delineation of regions of interest.

**UN-SAM.** UN-SAM (Chen et al., 2025b) introduces a domain-adaptive self-prompting framework for nuclei segmentation. To remove the manual annotations, it employs a self-prompt generation module to produce high-quality segmentation hints automatically. It further strengthens cross-domain generalization by combining shared representations with domain-specific adaptations, allowing robust performance on heterogeneous nuclei images.

**Med-SA.** Medical SAM Adapter (Med-SA) (Wu et al., 2025) extends SAM to medical imaging via parameter-efficient fine-tuning, updating only about 2% of its parameters. It introduces two key components: SD-Trans, which adapts SAM to 3D medical data, and the Hyper-Prompting Adapter (HyP-Adpt), which conditions the model on user-provided prompts for interactive segmentation. Med-SA supports both point clicks and bounding box prompts for prediction.

### B.5 FEATURE EXTRACTION BACKBONES

**UNI, UNI2.** UNI (Chen et al., 2024) is a pathology-specific vision encoder built with self-supervised pretraining on a large scale. More than 100 million image tiles drawn from around 100,000 diagnostic slides were used. It demonstrates strong transfer across a wide range of pathology tasks, particularly excelling in settings involving rare or underrepresented cancers. UNI-2 enlarges the pretraining corpus to over 200 million H&E and immunohistochemistry (IHC) images from more than 350,000 slides. By incorporating greater scale and modality diversity, UNI-2 further improves generalization across diagnostic tasks. It provides the community with an openly available resource for developing and benchmarking computational pathology models.

**Virchow, Virchow2.** The Virchow model family (Zimmermann et al., 2024; Vorontsov et al., 2024) represents an effort to establish foundation encoders tailored to digital pathology. Virchow was trained in a self-supervised manner on a vast collection of histopathology tiles obtained from millions of whole-slide images (WSIs). This large-scale pretraining equips the model with a broad awareness of tissue architecture and cellular morphology, which can then be transferred to downstream tasks such as cancer subtyping, outcome prediction, or biomarker discovery. Virchow2 extends this approach by further scaling the training corpus to more than 3.1 million WSIs, positioning it among the largest pathology encoders. Beyond its scale, Virchow2 can also serve as a frozen feature extractor for efficient pipeline integration or be fine-tuned to maximize performance on task-specific datasets.

**H-optimus-1.** H-optimus-1 (Bioptimus, 2025) is a 1.1B-parameter vision transformer developed by Bioptimus. It was pretrained with self-supervised learning on billions of histology tiles from over one million slides. It produces rich patch embeddings that capture complex spatial and structural relationships, supporting downstream tasks such as survival analysis, tissue classification, and segmentation.

**DINOv2, DINOv3.** DINOv2 (Oquab et al., 2024) is a self-supervised vision transformer pre-trained on a curated collection of 142 million images and is designed to produce general-purpose visual features comparable to those learned by weakly or fully supervised methods. Its representations transfer effectively across domains, enabling broad use without task-specific fine-tuning. DINOv3 (Siméoni et al., 2025) extends DINOv2 by scaling pretraining to over a billion images and models with billions of parameters. It further introduces mechanisms to stabilize training and preserve dense feature quality. With additional distillation into smaller, efficient variants, it achieves superior performance on both recognition and dense prediction tasks, underscoring the role of scale-aware design in vision foundation models.

**ViT-l/16.** The Vision Transformer (ViT) family adapts the Transformer architecture to images by treating fixed-size patches as tokens. Among its configurations, ViT-l/16 (Large, $16 \times 16$ patch size) (Dosovitskiy et al., 2020) has become a widely adopted backbone due to its balance of scale and granularity. It consists of 24 Transformer layers with a hidden dimension of 1024 and roughly 307M parameters. Pretraining on large-scale datasets demonstrates that ViT-l/16 gains substantial improvements from scaling. Its $16 \times 16$ patch size ensures a manageable sequence length while preserving sufficient spatial resolution, making it a widely used backbone and a standard reference model for feature extraction in vision transformers.

### B.6 ADDITIONAL ALGORITHMS

**Watershed algorithm.** The watershed algorithm (Vincent & Soille, 1991) is a classical segmentation approach widely employed for delineating touching or overlapping objects. It operates on the analogy of a topographic surface, where the grayscale image is interpreted as an elevation map. Pixels of lower intensity correspond to basins (valleys), whereas higher intensity values represent ridges (peaks). By conceptually flooding this landscape, water initially fills the basins, and as the level rises, neighboring catchment areas begin to merge. To prevent such mergers, separating boundaries are introduced at the points of convergence, thereby yielding distinct object regions. In practice, the algorithm is driven by the inverse of the distance transform, with local maxima serving as initial seeds or markers that guide the flooding process. Within our framework, the centroid of each resulting watershed region is used as the positive point prompt for prompt-based segmentation.

**Otsu's thresholding.** Otsu's thresholding (Otsu, 1979) is a global image binarization technique that selects the threshold $t^\star$ by minimizing the intra-class variance of foreground and background pixels, which is equivalent to maximizing the between-class variance.

Suppose the image histogram has $L$ gray levels, with normalized probabilities $p_i$ for each level $i$. For a given threshold $t$, the images are divided into two classes: class 0 with gray level $1, \cdots, t$ with probability $w_0(t) = \sum_{i=1}^{t} p_i$ and mean $\theta_0(t)$, class 1 with gray level $t+1 \cdots, L$ with probability $w_1(t) = \sum_{i=t+1}^{L} p_i$ and mean $\theta_1(t)$. The total mean is $\theta_T = \sum_{i=1}^{L} i p_i$. The between-class variance is:

$$\sigma^2(t) = w_0(t)(\theta_0(t) - \theta_T)^2 + w_1(t)(\theta_1(t) - \theta_T)^2$$

Otsu's method chooses the threshold:

$$t^\star = \arg \max_t \sigma^2(t)$$

which maximizes the separation between the two classes. This makes the method non-parametric and unsupervised, requiring no prior knowledge about the image content for foreground–background separation.

## C   ADDITIONAL QUALITATIVE RESULTS

In this section, we provide the qualitative visualizations of segmentation results from our SPROUT pipeline across three datasets (Section C.1), complementing the analysis in Figure 4. For comparison, we also include results from semantic and instance segmentation models developed for natural images in Section C.2. It helps illustrate the gap between natural and pathological domains and highlights the unique challenges of nuclear image segmentation discussed in Introduction (Section 1).

### C.1   SEGMENTATION RESULTS ACROSS DATASETS

As illustrated in Figure 10, 11, and 12, SPROUT delivers accurate nuclear segmentation across diverse datasets. The method demonstrates robustness under varying cellular organizations, from crowded fields containing thousands of nuclei to extremely sparse images with only a few. It performs well even when nuclei are small, densely packed, or exhibit subtle color contrasts with surrounding tissues. Moreover, SPROUT adapts seamlessly to datasets acquired with different staining protocols and image formats, including those with non-standard channels, consistently producing reliable delineations. Note that both predicted and ground-truth instance masks are displayed with randomly assigned colors, which serve only to differentiate instances for better visualization. Identical colors do not imply correspondence across cells or categories.

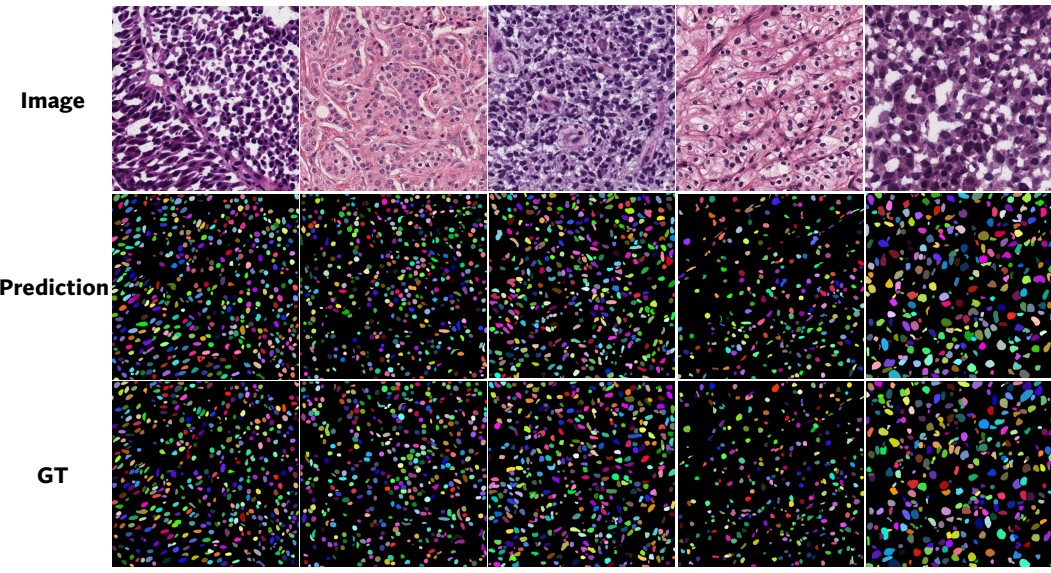

Figure 10: **Visualization of segmentation results on MoNuSeg.** Our method accurately distinguishes nuclear regions from surrounding white and stained tissues, and maintains robustness in dense and small nucleus scenarios. Each image may contain thousands of nuclei, and the SPROUT consistently identifies and segments them with high fidelity.

### C.2   COMPARISON WITH NATURAL IMAGE SEGMENTATION MODELS

To assess the intrinsic difference between segmentation models developed for natural images and those tailored to pathological images, we apply annotation-free natural image models to nuclear segmentation. Specifically, we choose SAM (Kirillov et al., 2023), MaskDINO (Li et al., 2023), and CutLER (Wang et al., 2023) for instance segmentation and Bridge the Points (Zhang et al., 2024a) for one-shot semantic segmentation. The detailed description of SAM is provided in Appendix Section B.4, while the remaining models are summarized below. The corresponding segmentation results are also presented for comparison.

**SAM.**   As shown in Figure 13, SAM auto-prompting can identify part of the nuclei but fails to reliably distinguish foreground from background. Large tissue regions are often misclassified as targets, which suppresses small-cell predictions either due to lower SAM scores or the lack of point

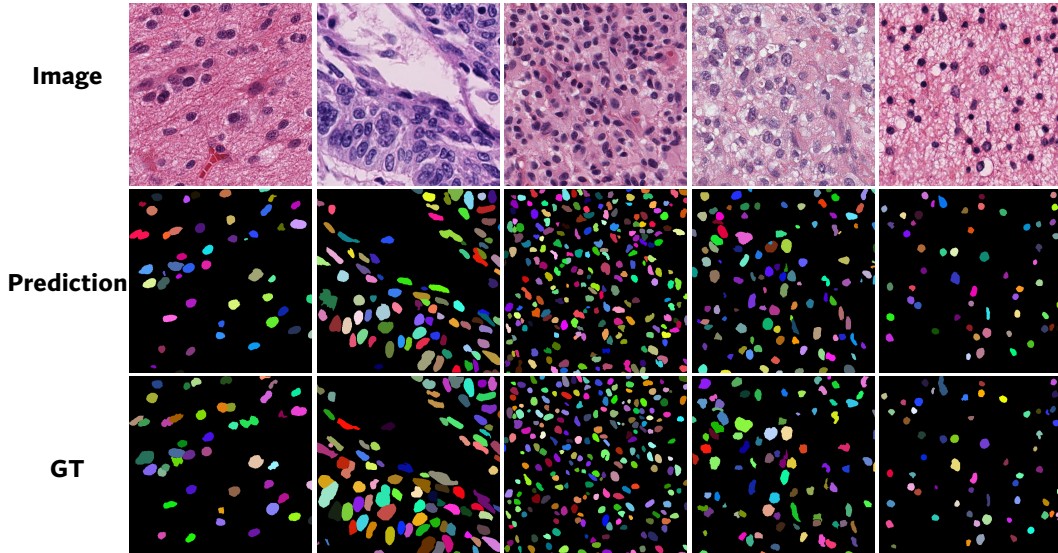

Figure 11: **Visualization of segmentation results on CPM17.** SPROUT effectively segments target nuclei in small-scale images when the input exhibits clear foreground–background separation, lightly separated nuclear regions, or tissue and nuclei with similar purple hues that make background discrimination challenging.

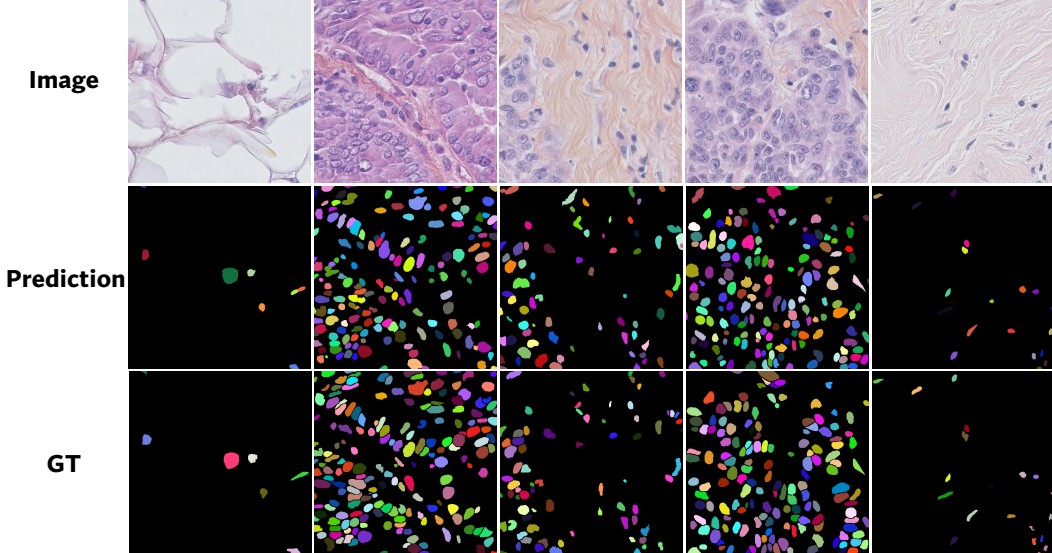

Figure 12: **Visualization of segmentation results on TNBC.** Our method remains effective on the TNBC dataset collected under different protocols with an additional transparency channel beyond standard RGB. It performs robustly in both sparse settings with only dozens of nuclei and highly crowded cases with hundreds, accurately distinguishing nuclear regions with minimal omission.

prompts generated on a coarse grid. In addition, without negative prompts, the masks frequently exhibit over-segmentation, reflecting the strong similarity between nuclear regions and surrounding background textures.

**Mask DINO.** Mask DINO extends the transformer-based object detector DINO by adding a parallel mask prediction branch, thereby unifying detection with instance, semantic, and panoptic segmentation through shared query embeddings for bounding box regression and mask generation.

Pretrained on large-scale detection and segmentation datasets, it achieves strong performance on natural image benchmarks.

However, as shown in Figure 14, its performance on MoNuSeg is limited. The top predictions frequently overlap on large tissue regions rather than capturing individual nuclei. This behavior suggests a tendency to segment regions of homogeneous texture or color areas, while failing to delineate fine-grained nuclear structures.

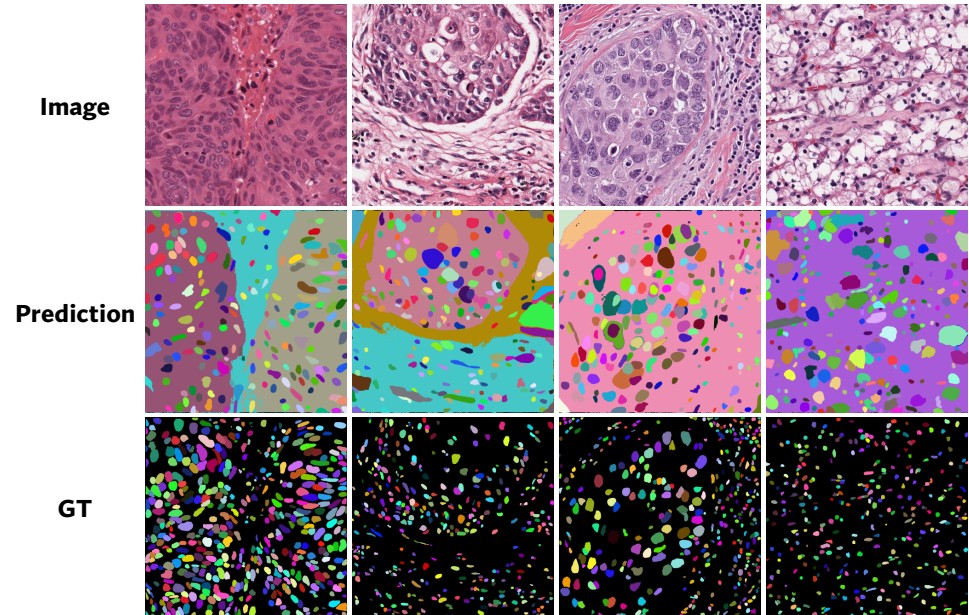

Figure 13: **Visualization of instance segmentation results from SAM auto-prompting.** SAM captures a subset of nuclei but struggles to separate nuclei from background tissues, leading to compromised segmentation.

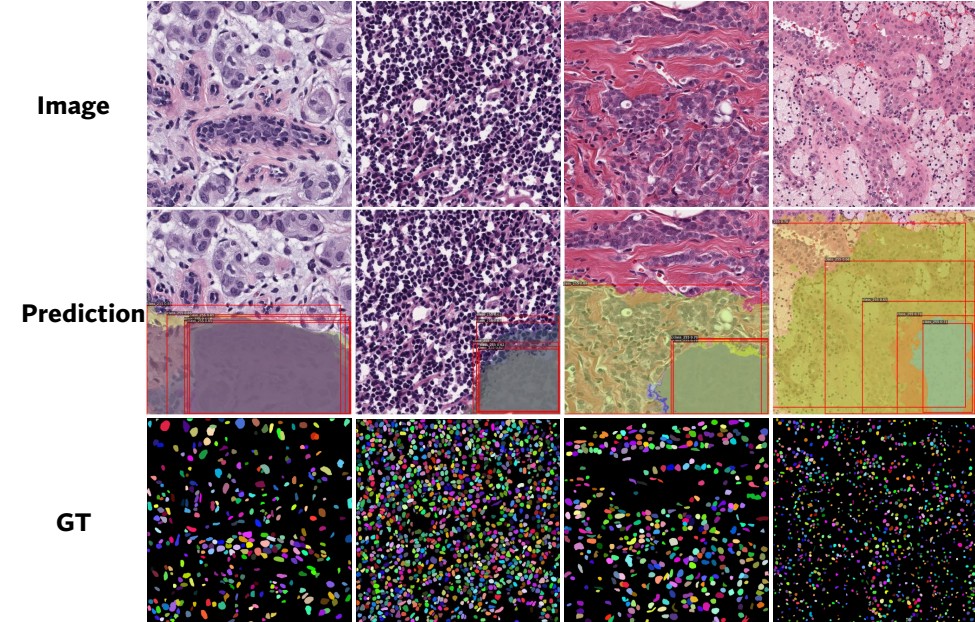

Figure 14: **Visualization of segmentation results from Mask DINO.** The model highlights broad tissue regions but fails to resolve individual nuclei.

**CutLER (Cut-and-Learn).** CutLER is a framework for unsupervised object detection and instance segmentation. It builds on MaskCut, which generates coarse object masks from self-supervised Vision Transformer features, and then refines them through detector training with robust loss dropping and iterative self-training. Operating entirely without human annotations, it provides a strong baseline for natural image segmentation.

In our experiments, only the MaskCut stage is applied to produce instance masks, without training the entire network. Because its affinity matrix relies on feature similarity, which is often weakly distinguishable in pathological images, CutLER primarily segments regions with sharp boundary contrasts but is unable to identify small-scale nuclei, as illustrated in Figure 15. Additionally, many nuclear regions remain undetected due to feature homogeneity.

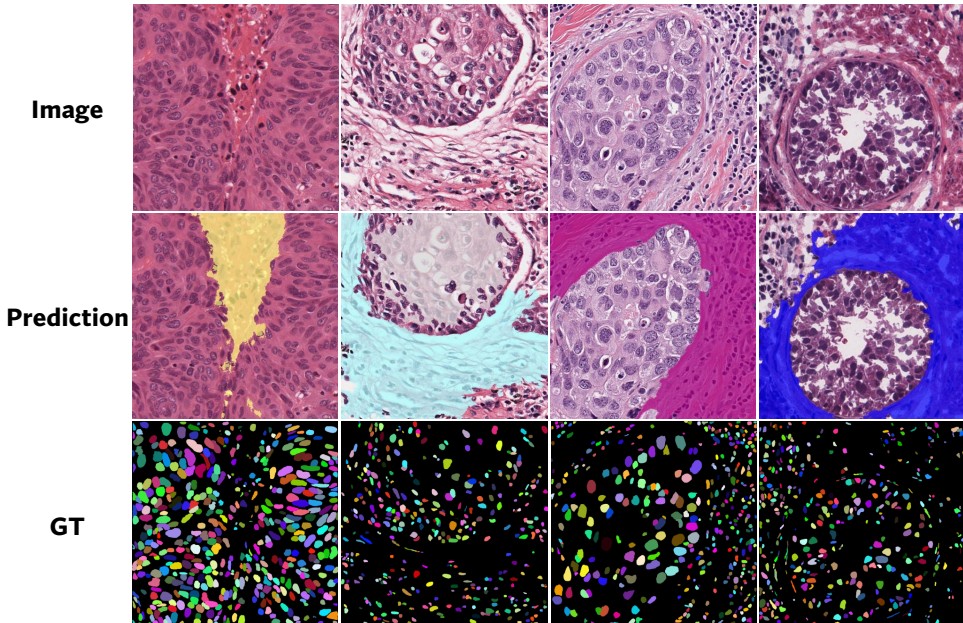

Figure 15: **Visualization of segmentation results from CutLER.** The model segments coarse regions with boundary changes but fails to capture fine-grained nuclear structures. The colored areas denote the produced masks.

**Bridge the Points.** Bridge the Points is a graph-based extension of SAM designed for few-shot semantic segmentation. It automatically selects informative prompts and aligns them with mask granularity through graph connectivity, reducing reliance on hyperparameters and redundant mask refinement. This makes it suitable for low-data settings and cross-domain generalization.

In our experiments, semantic masks from reference images were provided together with random target images in a one-shot configuration. As shown in Figure 16, the method improves the delineation of multiple smaller regions compared to other semantic approaches but still operates largely at the tissue level and fails to resolve nuclei at the instance level. Alongside the results from the instance segmentation baselines, this highlights the limitations of directly applying natural image models to the pathological domain.

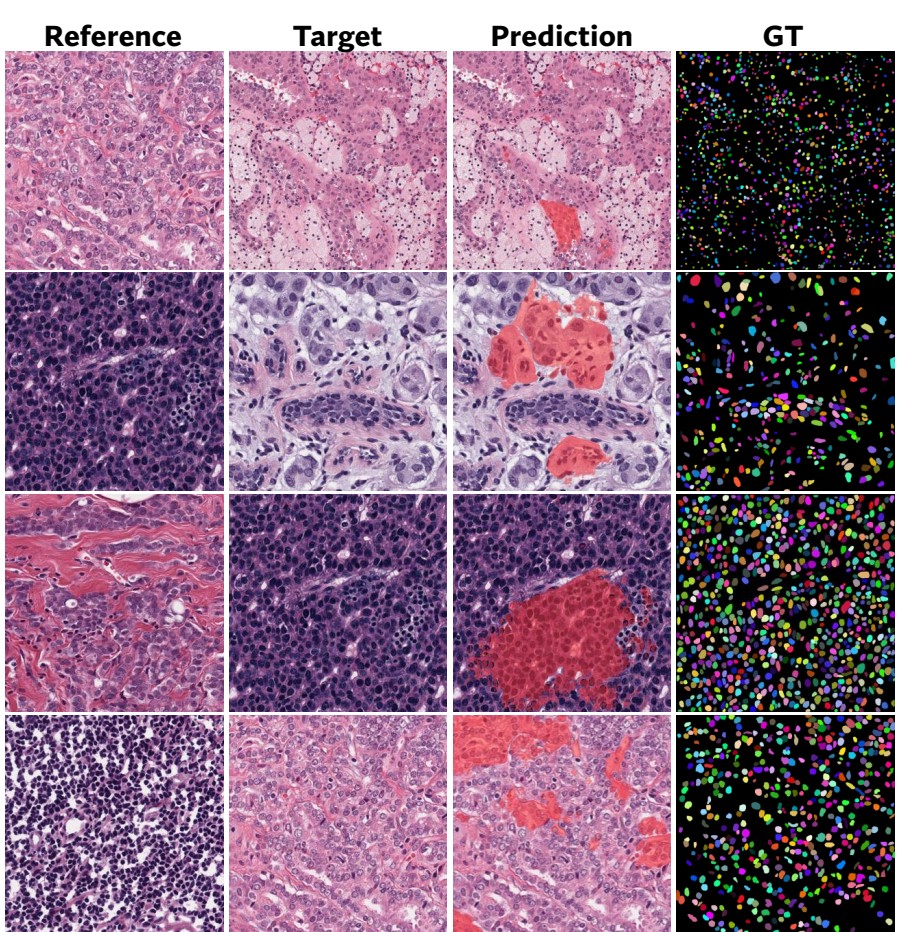

Figure 16: **Visualization of semantic segmentation results from Bridge the Points.** Red areas represent the segmented regions.

## D  ADDITIONAL ABLATION STUDY

In this section, we report comprehensive segmentation results on MoNuSeg, CPM17, and TNBC across all feature extraction backbones (Section D.1). The complete sets of evaluation metrics, together with additional ablation studies on point generation and post-processing, are provided in Section D.2 and Section D.3, respectively.

### D.1  FEATURE EXTRACTORS ACROSS DATASETS

We present detailed segmentation results on the MoNuSeg, CPM17, and TNBC datasets across different backbones and patch sizes (Table 2–10). The best results are highlighted in **bold**, and the second-best are underlined. These results confirm the effectiveness of our self-reference strategy, showing consistent generalization and comparable performance between natural-image-based and pathology-specific backbones. Unless otherwise specified, all subsequent predictions are obtained using the large SAM model with a patch size of $512 \times 512$ with an overlap ratio of $0.5$. Soft NMS is employed as the post-processing method.

Table 2: **Segmentation results of AJI and Dice with different backbones on the MoNuSeg dataset.**

| Backbones | AJI | | | | | Dice | | | | |
|---|---|---|---|---|---|---|---|---|---|---|
| | 64 | 128 | 256 | 512 | 1024 | 64 | 128 | 256 | 512 | 1024 |
| UNI | 0.580 | 0.593 | 0.606 | 0.593 | 0.599 | 0.772 | **0.795** | 0.787 | 0.766 | 0.778 |
| UNI2 | 0.582 | 0.615 | 0.602 | 0.577 | 0.576 | 0.774 | 0.780 | 0.777 | 0.760 | 0.761 |
| Virchow | 0.590 | 0.613 | 0.621 | 0.575 | 0.579 | 0.764 | 0.787 | 0.780 | 0.759 | 0.762 |
| Virchow2 | 0.614 | **0.622** | 0.604 | 0.581 | 0.575 | 0.770 | **0.795** | 0.776 | 0.763 | 0.759 |
| H-optimus-1 | 0.582 | 0.599 | 0.577 | 0.568 | 0.573 | 0.772 | 0.785 | 0.781 | 0.751 | 0.752 |
| ViT-l/16 | 0.592 | 0.606 | 0.596 | 0.569 | 0.570 | 0.764 | 0.784 | 0.768 | 0.760 | 0.758 |
| DINOv2 | 0.598 | 0.609 | 0.598 | 0.582 | 0.572 | 0.770 | 0.784 | 0.777 | 0.764 | 0.756 |
| DINOv3 | 0.612 | 0.617 | 0.612 | 0.596 | 0.589 | 0.779 | 0.789 | 0.780 | 0.766 | 0.773 |

Table 3: **Segmentation results of DQ and SQ with different backbones on the MoNuSeg dataset.**

| Backbones | DQ | | | | | SQ | | | | |
|---|---|---|---|---|---|---|---|---|---|---|
| | 64 | 128 | 256 | 512 | 1024 | 64 | 128 | 256 | 512 | 1024 |
| UNI | 0.752 | 0.805 | 0.793 | 0.774 | 0.772 | 0.730 | 0.731 | 0.733 | 0.729 | 0.723 |
| UNI2 | 0.751 | 0.803 | 0.792 | 0.762 | 0.769 | 0.732 | 0.733 | 0.733 | 0.730 | 0.726 |
| Virchow | 0.749 | 0.809 | 0.792 | 0.756 | 0.770 | 0.733 | 0.732 | 0.732 | 0.729 | 0.727 |
| Virchow2 | 0.785 | **0.817** | 0.803 | 0.770 | 0.762 | 0.734 | 0.736 | 0.733 | 0.731 | 0.728 |
| H-optimus-1 | 0.767 | 0.796 | 0.795 | 0.762 | 0.763 | 0.733 | 0.733 | 0.735 | 0.729 | 0.726 |
| ViT-l/16 | 0.764 | 0.797 | 0.785 | 0.751 | 0.762 | **0.739** | 0.731 | 0.731 | 0.728 | 0.725 |
| DINOv2 | 0.764 | 0.796 | 0.795 | 0.767 | 0.763 | 0.730 | 0.731 | 0.732 | 0.730 | 0.726 |
| DINOv3 | 0.789 | 0.813 | **0.817** | 0.780 | 0.777 | 0.731 | 0.734 | 0.734 | 0.729 | 0.728 |

Table 4: **Segmentation PQ results with different backbones on the MoNuSeg Dataset.**

| Backbones | 64 | 128 | 256 | 512 | 1024 |
|---|---|---|---|---|---|
| UNI | 0.549 | 0.588 | 0.581 | 0.564 | 0.558 |
| UNI2 | 0.550 | 0.589 | 0.581 | 0.556 | 0.558 |
| Virchow | 0.549 | 0.592 | 0.580 | 0.551 | 0.560 |
| Virchow2 | 0.576 | **0.601** | 0.589 | 0.563 | 0.555 |
| H-optimus-1 | 0.562 | 0.583 | 0.584 | 0.555 | 0.554 |
| ViT-l/16 | 0.565 | 0.582 | 0.574 | 0.547 | 0.552 |
| DINOv2 | 0.558 | 0.582 | 0.582 | 0.560 | 0.554 |
| DINOv3 | 0.577 | 0.597 | 0.600 | 0.569 | 0.566 |

Table 5: **Segmentation results of AJI and Dice with different backbones on the CPM17 dataset.**

| Backbones | AJI | | | | Dice | | | |
|---|---|---|---|---|---|---|---|---|
| | 64 | 128 | 256 | 512 | 64 | 128 | 256 | 512 |
| UNI | 0.641 | 0.651 | 0.639 | 0.594 | 0.805 | 0.812 | 0.802 | 0.769 |
| UNI2 | 0.645 | 0.652 | 0.642 | 0.596 | 0.803 | 0.815 | 0.802 | 0.762 |
| Virchow | 0.643 | 0.649 | 0.646 | 0.599 | 0.805 | 0.812 | 0.807 | 0.769 |
| Virchow2 | 0.642 | 0.640 | 0.635 | 0.592 | 0.794 | 0.806 | 0.787 | 0.768 |
| H-optimus-1 | 0.651 | **0.662** | 0.643 | 0.580 | 0.804 | 0.818 | 0.795 | 0.765 |
| ViT-l/16 | 0.649 | 0.654 | 0.642 | 0.583 | 0.809 | 0.810 | 0.804 | 0.759 |
| DINOv2 | 0.643 | 0.649 | 0.637 | 0.584 | 0.805 | 0.809 | 0.800 | 0.759 |
| DINOv3 | 0.648 | 0.657 | 0.637 | 0.595 | 0.805 | **0.821** | 0.799 | 0.766 |

Table 6: **Segmentation results of DQ and SQ with different backbones on the CPM17 dataset.**

| Backbones | DQ | | | | SQ | | | |
|---|---|---|---|---|---|---|---|---|
| | 64 | 128 | 256 | 512 | 64 | 128 | 256 | 512 |
| UNI | 0.777 | 0.787 | 0.787 | 0.739 | 0.768 | 0.770 | 0.771 | 0.763 |
| UNI2 | 0.771 | 0.786 | 0.780 | 0.739 | 0.763 | 0.768 | 0.770 | 0.763 |
| Virchow | 0.773 | 0.781 | 0.793 | 0.743 | 0.769 | 0.771 | 0.768 | 0.764 |
| Virchow2 | 0.767 | 0.780 | 0.783 | 0.738 | 0.768 | 0.769 | 0.770 | 0.768 |
| H-optimus-1 | 0.774 | **0.796** | 0.776 | 0.734 | 0.768 | **0.774** | 0.770 | 0.763 |
| ViT-l/16 | 0.784 | 0.789 | 0.787 | 0.725 | 0.767 | 0.769 | 0.772 | 0.761 |
| DINOv2 | 0.783 | 0.789 | 0.780 | 0.726 | 0.768 | 0.767 | 0.771 | 0.766 |
| DINOv3 | 0.788 | 0.794 | 0.789 | 0.747 | 0.768 | 0.770 | 0.772 | 0.766 |

Table 7: **Segmentation PQ results with different backbones on the CPM17 Dataset.**

| Backbones | 64 | 128 | 256 | 512 |
|---|---|---|---|---|
| UNI | 0.597 | 0.606 | 0.606 | 0.564 |
| UNI2 | 0.588 | 0.604 | 0.601 | 0.564 |
| Virchow | 0.594 | 0.602 | 0.609 | 0.568 |
| Virchow2 | 0.589 | 0.600 | 0.602 | 0.567 |
| H-optimus-1 | 0.594 | **0.616** | 0.599 | 0.560 |
| ViT-l/16 | 0.601 | 0.607 | 0.608 | 0.552 |
| DINOv2 | 0.601 | 0.605 | 0.604 | 0.556 |
| DINOv3 | 0.605 | 0.611 | 0.609 | 0.572 |

Table 8: **Segmentation results of AJI and Dice with different backbones on the TNBC dataset.**

| Backbones | AJI | | | | Dice | | | |
|---|---|---|---|---|---|---|---|---|
| | 64 | 128 | 256 | 512 | 64 | 128 | 256 | 512 |
| UNI | 0.601 | 0.600 | 0.587 | 0.566 | 0.767 | 0.773 | 0.761 | 0.746 |
| UNI2 | 0.602 | 0.593 | 0.587 | 0.577 | 0.761 | 0.765 | 0.761 | 0.754 |
| Virchow | 0.595 | 0.605 | 0.593 | 0.568 | 0.767 | 0.775 | 0.766 | 0.747 |
| Virchow2 | 0.592 | 0.599 | 0.575 | 0.565 | 0.767 | 0.769 | 0.752 | 0.742 |
| H-optimus-1 | 0.603 | 0.600 | 0.576 | 0.574 | 0.768 | **0.780** | 0.753 | 0.748 |
| ViT-l/16 | **0.609** | 0.604 | 0.594 | 0.570 | 0.774 | 0.766 | 0.766 | 0.746 |
| DINOv2 | 0.606 | 0.596 | 0.581 | 0.567 | 0.772 | 0.766 | 0.755 | 0.747 |
| DINOv3 | 0.595 | 0.576 | 0.580 | 0.576 | 0.761 | 0.759 | 0.756 | 0.754 |

Table 9: **Segmentation results of DQ and SQ with different backbones on the TNBC dataset.**

| Backbones | DQ | | | | SQ | | | |
|---|---|---|---|---|---|---|---|---|
| | 64 | 128 | 256 | 512 | 64 | 128 | 256 | 512 |
| UNI | 0.783 | 0.788 | 0.782 | 0.766 | 0.785 | _0.788_ | 0.787 | 0.781 |
| UNI2 | 0.780 | 0.790 | 0.787 | 0.767 | **0.789** | 0.787 | 0.786 | 0.782 |
| Virchow | 0.780 | **0.796** | 0.794 | 0.767 | 0.785 | 0.787 | 0.784 | 0.781 |
| Virchow2 | 0.792 | _0.795_ | 0.776 | 0.763 | 0.786 | 0.785 | 0.786 | 0.784 |
| H-optimus-1 | 0.785 | 0.794 | 0.774 | 0.775 | 0.787 | 0.786 | 0.783 | 0.779 |
| ViT-l/16 | 0.794 | 0.793 | 0.788 | 0.768 | 0.782 | 0.785 | 0.785 | 0.781 |
| DINOv2 | 0.794 | **0.796** | 0.781 | 0.772 | 0.784 | 0.787 | 0.785 | 0.778 |
| DINOv3 | 0.784 | 0.783 | 0.771 | 0.775 | _0.788_ | 0.785 | 0.785 | 0.781 |

Table 10: **Segmentation PQ results with different backbones on the TNBC Dataset.**

| Backbones | 64 | 128 | 256 | 512 |
|---|---|---|---|---|
| UNI | 0.615 | 0.621 | 0.615 | 0.598 |
| UNI2 | 0.615 | 0.622 | 0.619 | 0.600 |
| Virchow | 0.612 | **0.626** | 0.622 | 0.599 |
| Virchow2 | 0.623 | _0.624_ | 0.610 | 0.598 |
| H-optimus-1 | 0.618 | _0.624_ | 0.606 | 0.604 |
| ViT-l/16 | 0.621 | 0.623 | 0.619 | 0.600 |
| DINOv2 | 0.622 | **0.626** | 0.613 | 0.601 |
| DINOv3 | 0.618 | 0.615 | 0.605 | 0.605 |

## D.2 POINT QUALITY AND GENERATION

We report the detailed quantitative metrics corresponding to the ablation results presented in Figure 7 in Table 11. These results provide a comprehensive view of how each component contributes to the overall segmentation performance.

Table 11: **Comprehensive ablation results of point generation on MoNuSeg.** Partial denotes POT-Scan, while balanced refers to standard OT. The baseline corresponds to SAM auto-prompting.

| Color Prior Mask | Similarity Mapping | OT | AJI | PQ | DQ | SQ | Dice |
|---|---|---|---|---|---|---|---|
| ✗ | ✗ | ✗ | 0.061 | 0.262 | 0.384 | 0.752 | 0.353 |
| Otsu | ✗ | ✗ | 0.527 | 0.468 | 0.634 | 0.738 | 0.725 |
| Otsu | ✓ | ✗ | 0.532 | 0.468 | 0.636 | 0.736 | 0.728 |
| Otsu | ✓ | balanced | 0.545 | 0.474 | 0.647 | 0.732 | 0.742 |
| Otsu | ✓ | partial | 0.579 | 0.485 | 0.661 | 0.733 | 0.767 |
| High-confidence | ✓ | ✗ | 0.543 | 0.479 | 0.652 | 0.735 | 0.738 |
| High-confidence | ✓ | balanced | 0.567 | 0.510 | 0.686 | 0.743 | 0.759 |
| High-confidence | ✓ | partial | 0.619 | 0.577 | 0.787 | 0.733 | 0.795 |

To evaluate the generated point prompts both quantitatively and qualitatively, we provide the visualization of the point spatial distribution overlaid with the images in Figure 17 and the direct accuracy of both positive and negative points across the three datasets in Table 12. Overall, both positive and negative prompts exhibit high precision, with negative points achieving particularly strong true positive rates by effectively avoiding non-cell areas. Positive points also demonstrate robust coverage since they reliably capture nuclei even in densely packed regions with small cells. For larger nuclei, multiple positive points are often assigned within a single object, further increasing the possibility of accurate segmentation. Only a minor fraction of very small or weakly stained nuclei may be underrepresented, reflecting inherent challenges in such pathological settings.

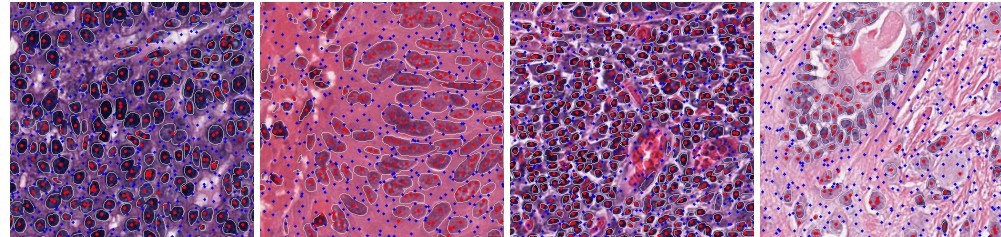

Figure 17: **Visualization of point prompts.** Positive prompts (red dots) and negative prompts (blue squares) are overlaid on the images to illustrate the quality of prompt selection, with white contours denoting the ground-truth cell boundaries. Images are cropped for clarity of visualization.

Table 12: **Evaluation of point prompt quality on different datasets.**

| Datasets | TP | TN | FP | FN |
|---|---|---|---|---|
| MoNuSeg | 0.849 | 0.151 | 0.943 | 0.057 |
| CPM17 | 0.873 | 0.127 | 0.956 | 0.044 |
| TNBC | 0.818 | 0.182 | 0.982 | 0.018 |

### D.3 SOFT NMS

In this section, we present a comprehensive evaluation of the proposed post-processing strategy. We report detailed performance metrics and ablation studies on the score and decay functions of soft NMS to validate the method. We further include visualizations and pseudo-code to illustrate the procedure.

As a complement to Figure 7b, Table 13 reports the detailed ablation results of the NMS strategy across all five evaluation metrics.

Table 13: **Detailed ablation results of post-processing strategy on MoNuSeg.**

| Containment Penalty | NMS | AJI | PQ | DQ | SQ | Dice |
|---|---|---|---|---|---|---|
| ✗ | hard | 0.553 | 0.523 | 0.712 | 0.735 | 0.743 |
| ✗ | soft | 0.573 | 0.529 | 0.726 | 0.729 | 0.765 |
| ✗ | hard+soft | 0.565 | 0.534 | 0.719 | 0.743 | 0.757 |
| ✓ | hard | 0.585 | 0.573 | 0.785 | 0.730 | 0.764 |
| ✓ | soft | 0.620 | 0.593 | 0.809 | 0.733 | 0.789 |
| ✓ | hard+soft | 0.605 | 0.576 | 0.788 | 0.731 | 0.778 |

To assess the necessity of the combined score strategy, we compare H-channel response alone, SAM confidence score alone, and their combination, with the raw predictions as the baseline. As shown in Table 14, applying soft NMS improves the overall segmentation performance compared with directly aggregating predictions. Both the H-channel response and the SAM score alone further enhance performance. The H-channel emphasizes heavily stained regions but tends to keep darker tissue areas. In contrast, the SAM score favors nuclei with clear boundaries, preserving large nuclear regions but failing to separate overlapping cells. The combined strategy effectively balances these complementary strengths and achieves the best overall performance.

We further investigate the effect of different score decay functions in soft NMS, including linear, polynomial, and exponential. Hard NMS is employed as the baseline. As summarized in Table 15, all variants of Soft-NMS consistently outperform hard NMS across multiple metrics, underscoring the advantage of retaining slight overlapping predictions with gradually decayed scores rather than discarding them. Among the tested functions, the exponential decay yields the best overall performance. This can be attributed to the fact that exponential decay provides a sharper penalty to highly overlapping instances while still preserving moderately overlapping candidates. It strikes a better balance between suppressing redundant detections and retaining true positives in the post-processing

procedure. The effectiveness of soft NMS with the exponential decay function is illustrated in Figure 18.

Table 14: **Ablation of score strategies for soft NMS on MoNuSeg.** Results highlight different focuses between H-channel response and SAM confidence. The combined approach provides the most balanced improvement in post-processing.

| Strategy | AJI | PQ | DQ | SQ | Dice |
|---|---|---|---|---|---|
| ✗ | 0.546 | 0.364 | 0.495 | 0.735 | 0.747 |
| $S'_H$ | 0.591 | 0.581 | 0.793 | 0.733 | 0.784 |
| $S_{SAM}$ | 0.587 | 0.583 | 0.792 | 0.732 | 0.781 |
| $S'_H + S_{SAM}$ | 0.619 | 0.593 | 0.809 | 0.734 | 0.792 |

Table 15: **Ablation of decay functions for soft NMS on MoNuSeg.** All soft NMS variants outperform hard NMS. Exponential decay achieves the best overall performance by more effectively penalizing heavily overlapping instances while preserving valid detections.

| Functions | AJI | PQ | DQ | SQ | Dice |
|---|---|---|---|---|---|
| hard | 0.587 | 0.549 | 0.761 | 0.721 | 0.764 |
| linear | 0.601 | 0.564 | 0.774 | 0.729 | 0.778 |
| polynomial | 0.599 | 0.568 | 0.778 | 0.731 | 0.780 |
| exponential | 0.614 | 0.577 | 0.787 | 0.733 | 0.787 |

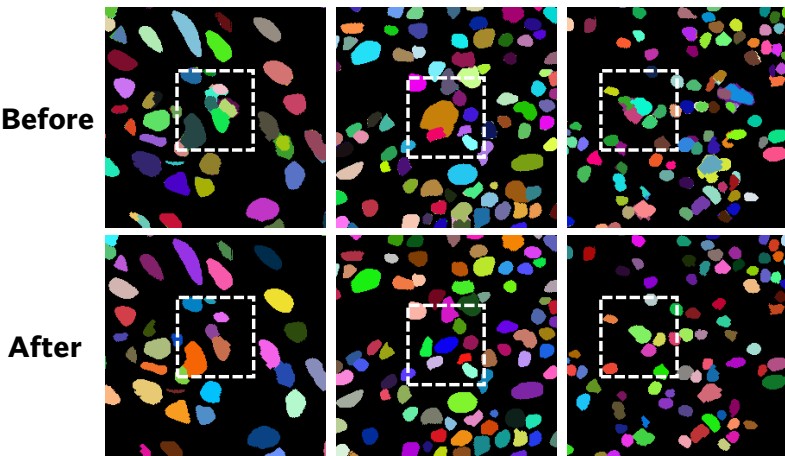

Figure 18: **Visualization of soft NMS post-processing.** The containment-aware variant effectively suppresses large predicted masks that contain smaller ones, leading to cleaner and more precise outputs. The highlighted regions show distinct differences.

Lastly, we provide the comprehensive pseudo-code of the containment-aware soft NMS in Alg.(3).

---

**Algorithm 3** Containment-aware Soft NMS

---

1: **Inputs:** Masks $\{M_i\}_1^n$, scores $\{S_i\}_1^n$
2: Exponential decay $f(x) = \exp(-x^2/\sigma)$ with $\sigma > 0$, penalty scale $\epsilon > 0$, score threshold $\tau$.
3:
4: $\mathcal{B} \leftarrow \emptyset$
5: $\mathcal{D} \leftarrow \{(M_i, S_i)\}_{i=1}^n$      ▷ working set of detections
6:
7: // Pre-decay containment penalty
8: **for** each $(M_i, S_i) \in \mathcal{D}$ **do**
9:      **if** $N_{\text{contained}}(M_i) > 1$ **then**
10:          $S_i \leftarrow S_i \cdot \big(1 - \tanh(\epsilon \cdot N_{\text{contained}}(M_i))\big)$
11:      **end if**
12: **end for**
13:
14: // Iterative Soft-NMS
15: **while** $\mathcal{D} \neq \emptyset$ **do**
16:      $(M_{\max}, S_{\max}) \leftarrow \arg\max_{(M,S) \in \mathcal{D}} S$
17:      $\mathcal{B} \leftarrow \mathcal{B} \cup \{M_{\max}\}$
18:      $\mathcal{D} \leftarrow \mathcal{D} \setminus \{(M_{\max}, S_{\max})\}$
19:      **for** each $(M_i, S_i) \in \mathcal{D}$ **do**
20:          $b_{\max} \leftarrow \text{bbox}(M_{\max}), \quad b_i \leftarrow \text{bbox}(M_i)$
21:          $u \leftarrow \text{IoU}(b_{\max}, b_i)$
22:          $S_i \leftarrow S_i \cdot f(u)$
23:          **if** $S_i < \tau$ **then**
24:              $\mathcal{D} \leftarrow \mathcal{D} \setminus \{(M_i, S_i)\}$      ▷ drop weak candidates
25:          **end if**
26:      **end for**
27: **end while**
28:
29: **return** $\mathcal{B}$

---

# E  ADDITIONAL SENSITIVITY ANALYSIS

## E.1  SAM EFFECT.

SAM is highly sensitive to the form of inputs. Naïve application often leads to suboptimal performance on pathology images due to densely packed structures, heterogeneous staining, and subtle nuclear boundaries. The strategies illustrated in the Figure 19 are designed to mitigate these challenges.

**Image size.**  As illustrated in Figure 19a, scaling controls the receptive field over which SAM attends. Without scaling, SAM can respond to large tissue regions, ignoring fine-grained nuclear detail. By splitting the image into small patches, the model is guided to treat individual nuclei as primary segmentation targets.

**Negative prompts.**  In histology images, nuclei are frequently embedded within complex backgrounds where clear boundaries are lacking. As shown in Figure 19b, negative prompts act as explicit counterexamples, guiding SAM to disregard surrounding tissue that mimics nuclear appearance. This strategy mitigates over-segmentation and enhances boundary precision.

**Positive prompt placement.**  Overlapping or touching nuclei pose a significant challenge, as SAM may merge them into a single mask in Figure 19c. Carefully placing positive prompts within each nucleus provides local anchors that enforce instance-level separation, enabling SAM to segment individual objects rather than clusters.

Together, these strategies systematically adapt SAM's general-purpose design to the demands of nuclear segmentation. They highlight that effective use of SAM in computational pathology depends not only on model capacity but also on thoughtful construction of inputs that reflect domain-specific image structure.

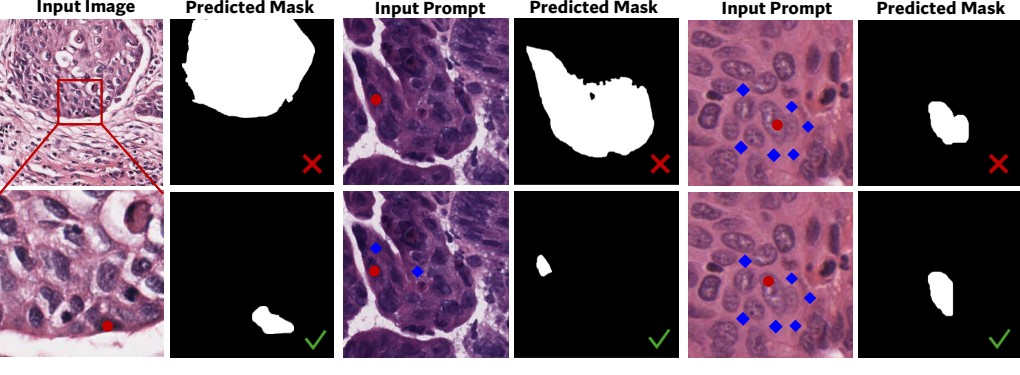

(a) Image Size        (b) Negative Points Presence        (c) Negative Points Presence

Figure 19: **Illustration of SAM effect.** (a) Appropriate image cropping enables SAM to focus on nuclei instead of broad tissue regions. (b) Incorporating negative prompts suppresses background responses for precise nuclear delineation. (c) The placement of positive prompts influences the separation of overlapping nuclei. Red dots denote positive prompts and blue squares denote negative prompts.

## F   THE USE OF LARGE LANGUAGE MODELS

In this work, Large Language Models (LLMs) are only used as an assistive tool for grammar checking and minor text editing. The role of LLMs is restricted to linguistic quality improvements. All ideation, experiment design, and analyses are the sole work of the authors.

