# OpenReview forum: "SPROUT: Training-free Nuclear Instance Segmentation with Automatic Prompting"
_ICLR.cc/2026/Conference — ICLR 2026 Conference Withdrawn Submission_

### Official Review · Reviewer_EXxU · 2025-10-28

**Soundness:** 2
**Presentation:** 3
**Contribution:** 1
**Rating:** 2
**Confidence:** 4

**Summary:**

The paper introduces SPROUT, a training-free method for H&E-specific cell instance segmentation. First, stain-guided self-reference maps select cell/background regions; features from a pretrained unsupervised semantic segmentation (e.g., DINOv2) are used to cluster similar cell/background pixels, which are subsequently refined via an optimal transport. Second, point prompts derived from the refined maps are fed to SAM to produce instance masks, followed by post-processing (e.g., watershed, NMS). Experiments on MoNuSeg, CPM-17, and TNBC report AJI/PQ, with ablations over parameter settings, prompt generation, and post-processing design choices.

**Strengths:**

1. The paper proposes a training- and annotation-free pipeline for cell instance segmentation by composing some components into a simple, reproducible workflow.
2. The study offers broad empirical coverage (MoNuSeg/CPM-17/TNBC; AJI/PQ) and systematic engineering ablations over hyperparameters (e.g., patch size), pretrained USS backbones, prompt generation, and post-processing.
3. The proposed SPROUT achieves comparable performance on CPM-17 and stronger performance on MoNuSeg than box/point-supervised baselines despite using zero supervision.

**Weaknesses:**

**1. Limited Novelty vs. COIN (SOTA for annotation-free cell instance segmentation).** SPROUT comprises the three-stage annotation-free framework: (i) stain-driven foreground/background propagation using unsupervised semantic segmentation (e.g., DINOv2), (ii) optimal-transport refinement to suppress pixel noise, and (iii) SAM-based instance mask generation using pseudo points. Apart from obtaining initial seeds via an H&E-specific prior and avoiding fine-tuning, three key components (USS/OT/SAM) and their ordering are functionally identical to COIN [1]. The paper does not articulate a principled distinction (e.g., why POT-Scan is theoretically or empirically superior to COIN’s OT stage under matched backbones, patching, prompting, and post-processing) nor provide a controlled head-to-head. As written, most  contributions and components used in SPROUT read as duplicative rather than novel.

**2. Insufficient Scope in Related Work.** In general segmentation (not only cell), USS/OT and their combinations [2, 3] have been widely explored, including hierarchical or region-aware schemes that explictly address imbalance and noise propagation. Recent papers [1, 4, 5] already integrate SAM with unsupervised and weak signals (e.g., text features from the pretrained CLIP). The paper should substantially widen its treatment of these directions and articulate component-level novelty beyond simple adaptation for cell segmentation (e.g., the proposed POT-Scan conceptually compared to prior OT approaches [1-3].

**3. Limited Modality Scope for Cell Segmentation (H&E Lock-in).** In contrast to cell segmentation methods [6, 7] that pursue cross-stain and cross-microscope generality, the proposed method relies on H&E-specific priors and currently offers insufficient evidence of transfer beyond H&E. It would strengthen the claim of impact to either demonstrate experimentally that non-H&E modalities can be integrated within the pipeline or articulate, with concrete justification and scope, why H&E constitutes the primary and practically relevant setup for the targeted applications.

**4. Heavy Heuristic Dependency.** Performance hinges on many tunables, such as high-confidence mask ratio, initial transported mass/weight, POT-Scan stride, K-means clustering, negative-point count, and patch-overlap ratio, yet sensitivity (min: 0.494, max: 0.621; Figure 9f). It would materially strengthen practicality to reduce or eliminate manual tuning by adopting non-parametric or data-driven selection rules in the spirit of COIN’s non-thresholding acceptance [1]. For example, overlap and prompt budgets derived from activation entropy or density, POT mass from distributional statistics, and K chosen via gap.

[1] COIN: Confidence Score-Guided Distillation for Annotation-Free Cell Segmentation, ICCV 2025.

[2] DHR: Dual Features-Driven Hierarchical Rebalancing in Inter- and Intra-Class Regions for Weakly-Supervised Semantic Segmentation, ECCV 2024.

[3] Point2Mask: Point-supervised Panoptic Segmentation via Optimal Transport, ICCV 2023.

[4] RADIOv2.5: Improved Baselines for Agglomerative Vision Foundation Models, CVPR 2025.

[5] Effective SAM Combination for Open-Vocabulary Semantic Segmentation, CVPR 2025.

[6] The Four Color Theorem for Cell Instance Segmentation, ICML 2025.

[7] The multimodality cell segmentation challenge: toward universal solutions, Nature Methods 2024.

**Questions:**

Most questions below directly target the core issues raised in Weaknesses.

Q1. Please compare the proposed method with COIN [1] conceptually. For each stage of the SPROUT's pipeline, state what is new, and show component swaps/ablations that isolate the effect of POS-Scan.

Q2. Either show a small result on at least one non-H&E stain or microscope, or explicitly limit claims to H&E and justify why this scope is practically important (e.g., clinical prevalence, workflow fit).

Q3. Can key manual settings be replaced with data-driven or non-parametric rules (e.g., entropy/density for prompt generation, distribution-based transported mass, confidence gap for K)?

Q4. Figure 5 shows similar AJI for pathology-specific backbones (UNI/UNI2/Virchow) and natural-image backbones (DINOv2/v3). Please explain why domain-specific pretraining offers little advantage here (e.g., effect of self-reference masks, prototype clustering, and patch scale). Under the same protocol, can you (i) swap the USS backbone for diffusion features and report AJI/PQ with runtime/memory (e.g., diffusion-based features as in [8, 9]), and (ii) test whether activation-derived prompting from diffusion attention can replace or complement stain-prior prompts?

Q5. Beyond the initial H&E prior, do POT-Scan and activation-based prompting (SAM) transfer to general or unsupervised instance segmentation? Please test these modules on standard baselines [10, 11] or provide a concrete analysis of feasibility and expected impact.

[8] Diffuse, Attend, and Segment: Unsupervised Zero-Shot Segmentation using Stable Diffusion, CVPR 2024.

[9] DiffCut: Catalyzing Zero-Shot Semantic Segmentation with Diffusion Features and Recursive Normalized Cut, NeurIPS 2024.

[10] Cut and Learn for Unsupervised Object Detection and Instance Segmentation, CVPR 2024.

[11] Scene-Centric Unsupervised Panoptic Segmentation, CVPR 2025.

---

### Official Review · Reviewer_eq4u · 2025-10-30

**Soundness:** 3
**Presentation:** 3
**Contribution:** 3
**Rating:** 4
**Confidence:** 5

**Summary:**

This paper targets an important topic for pathology image instance segmentation for cells, nuclei, tissue, etc. The authors introduce a prompt-based method named SPROUT, which leverages histology priors to mitigate domain gaps of annotation-free prompting to nuclear representations. In addition, this work included positive and negative prompts to the SAM-based model that enriched the method with negative priors.

**Strengths:**

- This paper is easy to follow, and the writing is clear and logical.
- The utilization of staining prior to is a good, as staining variations can be significant for HE slide images.
- The cost function of the OT solver is something new to measure global distance.

**Weaknesses:**

- Prompting a SAM model to pathology is a relatively old topic; the community has been exploring the nuclei segmentation a lot by implementing a SAM-based model with click prompts, bbx prompts or mask prompts. In this paper, the author can highlight more on what special challenges this study can solve, why we need another SAM variant for nuclear segmentation.
- What is a feature prototype here in the study, like a Global feature? Sounds like a vague term, the author can specify the prototype, or pattern referred to in the HE slide images.
- As mentioned, the SPROUT method is highlighted to capture staining features as H prior and E prior. How is this effective? How closely of these 2 clusters related? The author can show more visualization on this step
- How is optimal transport motivated here? Could a simpler metric, such as embedding distance or similarity distance, work?
- The implementation lacks some details on how the training-free model is derived; the authors can state more on the similarity mapping, - OT solver for the models.
- The datasets used are classic and small: MoNuSeg, CPM17, and TNBC.

**Questions:**

Questions and suggestions are associated with the weakness section. Thanks.

---

### Official Review · Reviewer_qEYc · 2025-10-31

**Soundness:** 2
**Presentation:** 3
**Contribution:** 2
**Rating:** 4
**Confidence:** 4

**Summary:**

This paper introduces SPROUT, a training-free framework for nuclear instance segmentation in histopathology images using the Segment Anything Model (SAM).
SPROUT eliminates the need for annotations or fine-tuning by combining:

Self-reference prompting — leveraging H&E stain priors to extract image-specific foreground/background features,
POT-Scan — a theoretically grounded partial optimal transport scheme to refine feature–prototype matching, and
Containment-aware NMS — to handle overlapping nuclei during mask refinement.

Across three datasets (MoNuSeg, CPM17, TNBC), SPROUT outperforms both fully supervised and weakly supervised baselines, achieving up to +8.2% AJI gain on MoNuSeg and comparable Dice scores to state-of-the-art methods — all without retraining or external annotations.

**Strengths:**

The partial OT formulation is mathematically rigorous, with clear derivations and proofs in the appendix.

The self-reference mechanism using stain priors is biologically plausible and empirically validated.

Experiments are extensive and well-controlled, with clear baselines (U-Net, MedSAM, UN-SAM, etc.) and ablations for each component (POT-Scan, prompt generation, NMS).

Robustness and sensitivity analyses are thorough — AJI variation <3% across hyperparameters demonstrates strong stability.

**Weaknesses:**

Reliance on SAM: Performance upper-bounded by SAM’s mask precision, especially for irregular or blurry nuclei boundaries.

Limited modality generalization: Evaluations are restricted to H&E-stained slides; generalization to other stains or modalities (e.g., immunohistochemistry) is untested.

Computational complexity: Although “training-free,” the pipeline still involves OT-based computations, clustering, and DenseCRF — runtime analysis beyond Fig. 6 would be valuable.

Lack of quantitative explainability metrics: The interpretability claims (e.g., via prototypes) could be better supported with formal measures.

Appendix-dependence: Many critical implementation details are relegated to appendices, making reproducibility harder without reading the supplement.

**Questions:**

Could SPROUT generalize to non-H&E stains (e.g., IHC, fluorescent images) or non-nuclear segmentation tasks?

How does partial OT behave when features exhibit extreme noise — does it risk discarding rare but valid nuclei?

Can SPROUT be integrated with smaller vision backbones for deployment on lower-resource pathology systems?

What is the computational overhead compared to simple heuristic prompting (e.g., Otsu or balanced OT)?

Is there potential for self-improving prompting, where the generated masks refine subsequent iterations?

---

### Official Review · Reviewer_EbcG · 2025-11-01

**Soundness:** 3
**Presentation:** 4
**Contribution:** 3
**Rating:** 6
**Confidence:** 3

**Summary:**

This paper presents SPROUT, a training- and annotation-free prompting framework for nuclear instance segmentation using SAM without fine-tuning. SPROUT generates automatic prompts through stain-based self-reference priors and refines predictions with containment-aware postprocessing. The method is evaluated on three public datasets and achieves consistent improvements over baseline across most metrics.

**Strengths:**

(1)  The core idea is interesting and practical — SPROUT effectively leverages SAM’s general segmentation ability without additional annotations or training, focusing instead on improving the quality of prompt generation.

(2) The paper provides comprehensive ablation studies and qualitative visualizations, which help validate the contribution of each component and clarify the design rationale.

**Weaknesses:**

The computational efficiency of the method is not clearly discussed. It would be useful to report runtime, GPU memory usage, or computational overhead compared to fine-tuned SAM or other training-free methods.

**Questions:**

From Table 11, the baseline SAM performs poorly across most metrics (e.g., Dice score ≈ 0.35), which seems unexpectedly low. Could the authors clarify why SAM performs so poorly in this setting? Additionally, the performance improves significantly after applying Otsu-based foreground–background separation, while subsequent modules contribute smaller incremental gains. Does this suggest that foreground separation is the main factor driving the improvement, and how do the other modules contribute beyond this step?

---

### Note · Authors · 2025-11-12

I have read and agree with the venue's withdrawal policy on behalf of myself and my co-authors.